# A neural network model of when to retrieve and encode episodic memories

Qihong Lu[1,2]*, Uri Hasson[1,2], Kenneth A Norman[1,2]

[1]Department of Psychology, Princeton University, Princeton, United States; [2]Princeton Neuroscience Institute, Princeton University, Princeton, United States

**Abstract** Recent human behavioral and neuroimaging results suggest that people are selective in when they encode and retrieve episodic memories. To explain these findings, we trained a memory-augmented neural network to use its episodic memory to support prediction of upcoming states in an environment where past situations sometimes reoccur. We found that the network learned to retrieve selectively as a function of several factors, including its uncertainty about the upcoming state. Additionally, we found that selectively encoding episodic memories at the end of an event (but not mid-event) led to better subsequent prediction performance. In all of these cases, the benefits of selective retrieval and encoding can be explained in terms of reducing the risk of retrieving irrelevant memories. Overall, these modeling results provide a resource-rational account of why episodic retrieval and encoding should be selective and lead to several testable predictions.

## Editor's evaluation

This paper addresses an important problem in control of episodic memory. This paper develops a computationally-based proposal about how semantic, working memory, and episodic memory systems might learn to interact so that stored episodic memories can optimally contribute to reconstruction of semantic memory for event sequences. This is an understudied area and this present work can make a major theoretical contribution to this domain with new predictions. The reviewers were positive about the contribution in review, and the revisions have clarified the model in a number of important ways, including through some additional simulation and analysis.

*For correspondence:
qlu@princeton.edu

## Introduction

In a natural setting, when should an intelligent agent encode and retrieve episodic memories? For example, suppose I am viewing the BBC television series *Sherlock*. Should I retrieve an episodic memory that I formed when I watched earlier parts of the show, and if so, when should I retrieve this memory? When should I encode information about the ongoing episode?

Although episodic memory is one of the most studied topics in cognitive psychology and cognitive neuroscience, the answers to these questions are still unclear, in large part because episodic memory research has traditionally focused on experiments using simple, well-controlled stimuli, where participants receive clear instructions about when to encode and retrieve. For example, a typical episodic memory experiment could ask participants to remember a set of random word-pairs; later on, given a word-cue, the participants need to report the associated word (*Kahana, 2012*). In this kind of word-pair experiment, the optimal timing for encoding and retrieval is clear: The participant should encode an episodic memory when they study a word-pair and retrieve the associate when they are prompted by a cue. Existing computational models of human memory have similarly focused on discretized list-learning paradigms like the (hypothetical) word-pair learning study described above – these models (e.g., see *Gillund and Shiffrin, 1984*; *Hasselmo and Wyble, 1997*; *Shiffrin and Steyvers, 1997*;

**eLife digest** The human brain can record snapshots of details from specific events – such as where and when the event took place – and retrieve this information later. Recalling these 'episodic memories' can help us gain a better understanding of our current surroundings and predict what will happen next.

Studies of episodic memory have typically involved observing volunteers while they perform simple, well-defined tasks, such as learning and recalling lists of random pairs of words. However, it is less clear how episodic memory works 'in the wild' when no one is quizzing us, and we are going about everyday activities.

Recently, researchers have started to study memory in more naturalistic situations, for example, while volunteers watch a movie. Here, Lu et al. have built a computational model that can predict when our brains store and retrieve episodic memories during these experiments.

The team gave the model a sequence of inputs corresponding to different stages of an event, and asked it to predict what was coming next. Intuitively, one might think that the best use of episodic memory would be to store and retrieve snapshots as frequently as possible. However, Lu et al. found that the model performed best when it was more selective – that is, preferentially storing episodic memories at the end of events and waiting to recover them until there was a gap in the model's understanding of the current situation. This strategy may help the brain to avoid retrieving irrelevant memories that might (in turn) result in the brain making incorrect predictions with negative outcomes.

This model makes it possible for researchers to predict when the brain may store and retrieve episodic memories in a particular experiment. Lu et al. have openly shared the code for the model so that other researchers will be able to use it in their studies to understand how the brain uses episodic memory in everyday situations.

---

*Sederberg et al., 2008*; *Howard and Kahana, 2002*; *Norman and O'Reilly, 2003*; *Polyn et al., 2009*; *Cox and Criss, 2020*; for reviews, see *Norman et al., 2008*; *Criss and Howard, 2015*) are primarily designed to answer questions about what happens as a result of a particular sequence of encoding and retrieval trials, not questions about when encoding and retrieval should occur in the first place.

Recently, there has been increasing interest in using naturalistic stimuli such as movies or audio narratives in psychological experiments, to complement results from traditional experiments using simple and well-controlled stimuli (*Sonkusare et al., 2019*; *Nastase et al., 2020*). These experiments have the potential to shed light on when encoding and retrieval take place during event perception in a naturalistic context, where no one is explicitly instructing participants about how to use episodic memory. These studies have found evidence that episodic encoding and retrieval occur *selectively* over time. For example, results from fMRI studies suggest that episodic encoding occurs preferentially at the ends of events (*Baldassano et al., 2017*; *Ben-Yakov et al., 2013*; *Ben-Yakov and Henson, 2018*; *Reagh et al., 2020*), and episodic retrieval happens preferentially when people are uncertain about the ongoing situation (*Chen et al., 2016*). Selectivity effects can also be observed in the realm of more traditional list-learning studies – for example, there is extensive behavioral and neuroscientific evidence that stimuli that trigger strong prediction errors are preferentially encoded into episodic memory (for reviews, see *Frank and Kafkas, 2021a*; *Quent et al., 2021b*).

The goal of the present work is to develop a computational model that can account for *when episodic encoding and retrieval take place* in naturalistic situations; the model is meant to capture key features of neocortical-hippocampal interactions, as described below. We formalize the task of event processing by assuming that events involve sequences of states drawn from some underlying event schema, and that the agent's goal is to predict upcoming states. We then seek to identify policies for episodic encoding and retrieval by optimizing a neural network model on the event processing task. We analyze how the optimal policy changes under different environmental regimes, and how well this policy captures human behavioral and neuroimaging data. To the extent that they match, the model can be viewed as providing a *resource-rational* account of those findings (i.e., an explanation of how these encoding and retrieval policies arise as a joint adaptation to the constraints imposed by the human cognitive architecture and the constraints imposed by the task environment; *Griffiths et al., 2015*; *Lieder and Griffiths, 2019*; see also *Anderson and Schooler, 2000*; *Gershman, 2021*).

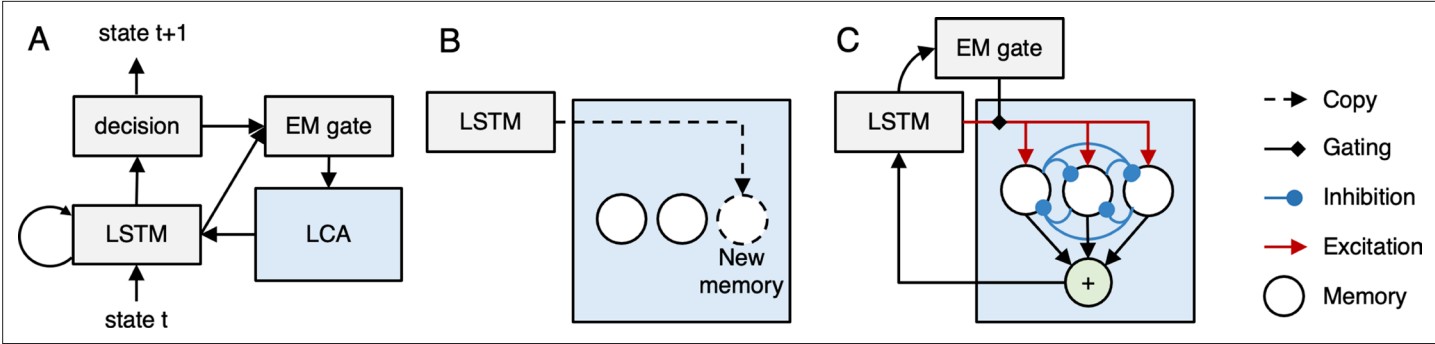

**Figure 1.** Neocortical-hippocampal Model. (**A**) At a given moment, the neocortical part of the model (shown in gray) observes the current state and predicts the upcoming state. It incorporates a Long Short Term Memory (LSTM; *Hochreiter and Schmidhuber, 1997*) network, which integrates information over time; the LSTM feeds into a non-linear decision layer. The LSTM and decision layers also project to an episodic memory (EM) gating layer that determines when episodic memories are retrieved (see part C of figure). The entire neocortical network is trained by an advantage-actor-critic (A2C) objective (*Mnih et al., 2016*) to optimize next-state prediction. (**B**) Episodic memory encoding involves copying the current hidden state and appending it to the list of memories stored in the episodic memory system (shown in blue), which is meant to correspond to hippocampus. (**C**) Episodic memory retrieval is implemented using a leaky competing accumulator model (LCA; *Usher and McClelland, 2001*) – each memory receives excitation proportional to its similarity to the current hidden state, and different memories compete with each other via lateral inhibition. The EM gate (whose value is set by the EM gate layer of the neocortical network) scales the level of excitation coming into the network. After a fixed number of time steps, an activation-weighted sum of all memories is added back to the cell state of the LSTM.

Overall, we find that the best-performing policies are selective in when encoding and retrieval take place, and that the types of selectivity identified by the model line up well with types of selectivity identified empirically. The key intuition behind these effects is that – while retrieving episodic memories can help us to predict upcoming states – there are risks to episodic retrieval: If you retrieve an irrelevant memory, you could make confident, wrong predictions that have negative consequences. The selective encoding and retrieval policies identified by the model help it to mitigate these risks while retaining the benefits of episodic memory. In the sections that follow, we describe our neocortical-hippocampal model, how we applied it to the tasks of interest, and the results of our simulations.

## Results

### A neural network model of neocortical-hippocampal interaction

Our modeling work leverages recent advances in memory-augmented neural networks (*Graves et al., 2016*; *Ritter et al., 2018*), deep reinforcement learning (*Mnih et al., 2016*; *Sutton and Barto, 2018*), and meta-learning (*Wang et al., 2018*; *Botvinick et al., 2019*) – these advances (collectively) make it possible for neural network models to *learn to use episodic memory* in the service of prediction (for an earlier precedent, see *Zilli and Hasselmo, 2008*).

Our model (*Figure 1A*) has two parts, which are meant to correspond to neocortex and hippocampus, and which collectively implement three key memory systems (working memory, semantic memory, and episodic memory). The neocortical part of the model incorporates a Long-Short-Term Memory module (LSTM; *Hochreiter and Schmidhuber, 1997*), which is a recurrent neural network (RNN) with gating mechanisms. In addition to the LSTM module, the neocortical network also incorporates a nonlinear decision layer (to assist with mapping inputs to next-state predictions) and an episodic memory (EM) gating layer, the function of which is described below. The LSTM module gives the neocortical network the ability to actively maintain and integrate information over time. For terminological convenience, we will refer to this active maintenance ability in the paper as 'working memory'. However, we should emphasize that – contrary to classic views of working memory (e.g., *Baddeley, 2000*) – our model does not have a working memory buffer that is set apart from other parts of the model that do stimulus processing; rather, active maintenance and integration are accomplished via recurrent activity in the parts of the model that are doing stimulus processing. In this respect, the architecture of our model fits with the *process memory* framework set forth by *Hasson et al., 2015*. In addition to this active maintenance ability, the connection weights of the neocortical network gradually extract regularities from the environment over time; this gradual learning of

regularities can be viewed as an implementation of semantic memory (*McClelland and Rumelhart, 1987*; *McClelland and Rogers, 2003*; *Rogers and McClelland, 2004*; *Saxe et al., 2019*).

The neocortical network is also connected to an episodic memory module (meant to simulate hippocampus) that stores snapshots of neocortical activity patterns (*Figure 1B*) and reinstates these patterns to the neocortical network; see the next section for more information on the model's encoding policy (i.e., when it stores snapshots). Episodic memory retrieval (*Figure 1C*) is implemented via a leaky competing accumulator process (LCA; *Usher and McClelland, 2001*; *Polyn et al., 2009*). In the LCA, memories compete to be retrieved according to how well they match the current state of the neocortical network, and the output of this competitive retrieval process is added back into the neocortical network. Crucially, the degree to which memories are activated during the retrieval process is multiplicatively gated by the EM gate layer of the neocortical network – this gives the neocortical network the ability to shape when episodic retrieval occurs (for more details on how EM works in the model, see the *Episodic retrieval* section in the Materials and methods).

The entire neocortical network (composed of the LSTM, decision, and EM gate layers) is trained via a reinforcement learning algorithm to optimize prediction of the next state given the current state as input; the trainable nature of the EM gate allows the network to learn a policy for when episodic memory retrieval should occur, in order to optimize next-state prediction. Specifically, we used a meta-learning procedure (*Wang et al., 2018*) whereby the model was trained repeatedly on all conditions of interest with modifiable neocortical weights (*meta-training*), before being evaluated in these conditions with neocortical weights frozen (*meta-testing*). This procedure captures the idea that neocortical weights only change gradually (*McClelland et al., 1995*), and thus are unlikely to be modified enough by one experience to support recall of unique aspects of that experience; as such, memory for these unique details depends critically on that information being held in working memory or episodic memory (for more details, see the *Model training and testing* section in the Materials and methods).

During meta-training, the model is rewarded for correct next-state predictions and punished for incorrect next-state predictions; we also gave the model the option of saying 'don't know' (instead of predicting a specific next state), in which case it receives zero reward. In the real world, there are often different costs associated with making commission errors (wrong predictions) and omission errors (not making a prediction). Having the 'don't know' option gives the model the freedom to choose whether it should make a specific prediction (thereby incurring the risk of making a commission error and receiving a penalty) or whether it should express uncertainty to avoid a possible penalty. Intuitively, this choice should depend on the environment. For example, if the penalty for misprediction is zero, the model should make a prediction even if it has high uncertainty about the upcoming state. In contrast, if the penalty for misprediction is high, the model should only make a prediction if it is certain about what would happen next. Practically speaking, the consequence of including the 'don't know' option is to induce the model to wait longer to retrieve episodic memories (see results below and also Appendix 5).

## Modeling the contribution of episodic memory to naturalistic event understanding

Our initial modeling target was a recent study by *Chen et al., 2016*, which explored the role of episodic memory in naturalistic event understanding. In this study, participants viewed an episode from the *Twilight Zone* television series. This episode was divided into two parts (part 1 and part 2). Participants in the recent memory (RM) condition viewed the two parts back-to-back; participants in the distant memory (DM) condition had a 1-day gap in between the two parts of this TV episode; participants in the no memory (NM) condition only watched the second part (*Chen et al., 2016*). In the RM condition, participants can build up a situation model – that is, a representation of the relevant features of the ongoing situation (*Richmond and Zacks, 2017*; *Stawarczyk et al., 2021*; *Zacks, 2020*; *Ranganath and Ritchey, 2012*) – during the first part of the movie and actively maintain it over time; all of that information is still actively represented at the start of part 2. By contrast, in the DM condition, a day has passed between part 1 and part 2, so participants are no longer actively maintaining the relevant situation model at the start of part 2.

Taken together, these conditions can be viewed as manipulating the *availability* of relevant episodic memories and also the *demand* for episodic retrieval. In the NM condition, at the start of part 2,

participants have gaps in their situation model (because they did not view part 1) and thus there is a strong demand to fill those gaps, to better understand what is going on; however, they do not have any relevant episodic memories available to fill those gaps. In the DM condition, because of the 1-day delay, participants also have gaps in their representation of the situation in working memory that need to be filled with information from part 1; however, unlike the NM participants, DM participants can meet this demand by retrieving information about part 1 that was stored in episodic memory. In the RM condition, like the DM condition, participants have relevant information about part 1 available in episodic memory (participants' experience in part 1 of the DM and RM conditions was identical, so presumably they stored the same episodic memories during part 1), but there is less of a demand to retrieve these episodic memories in the RM condition (because these participants were not interrupted, and thus these participants should have fewer gaps in their understanding of the situation). The comparison of the RM and DM conditions is thus a relatively pure manipulation of demand for episodic memory retrieval. If episodic memory retrieval is sensitive to the need to retrieve (i.e., whether there are gaps to fill in), then more retrieval should take place in the DM condition, but if episodic memory retrieval is automatic, retrieval should occur at similar levels in the RM and DM conditions. The results of the *Chen et al., 2016* study strongly support the former ('demand-sensitive') view of episodic retrieval. During the first two minutes of part 2, the researchers found strong hippocampal-neocortical activity coupling measured using inter-subject functional connectivity (ISFC; computed as functional connectivity across participants; *Simony et al., 2016*) for DM participants, while the level of coupling was much weaker for participants in the RM and NM conditions (*Chen et al., 2016*). Notably, neocortical regions that had a strong coupling with the hippocampus (in the DM condition) largely overlapped with the default mode network (DMN), which is believed to actively maintain a situation model (*Stawarczyk et al., 2021*). These results fit with the idea that more information is being communicated between hippocampus and neocortex in the DM ('high episodic memory demand') condition than in the RM ('low episodic memory demand') condition and the NM condition (where there are no relevant episodic memories to retrieve). This 'demand sensitive' view of episodic memory implies that neocortex can be strategic in how it calls upon the hippocampus to support event understanding, and it underlines the importance of the aforementioned goal of characterizing the policy for when retrieval should occur.

## Training environment

To simulate the task of event processing, we define an *event* as a sequence of states, sampled from an underlying graph that represents the *event schema*. **Figure 2A** shows a 'coffee shop visit' event schema graph with three time points; each time point has two possible states. Each instance of an event (here, each visit to the coffee shop) is associated with a *situation* – a collection of features set to particular values; importantly, the features of the current situation deterministically control the transitions between states within the event. For example, in **Figure 2A**, the value of the Weather situation feature (sunny or rainy) determines which of the Mood states is visited (happy or angry). That is to say, P(Mood = happy | Weather = sunny) = 1 and P(Mood = sad | Weather = rainy) = 1. At each time point, the model observes the value of a randomly selected feature of the current situation and responds to a query about which state will be visited next. In the example shown in **Figure 2A**, the agent first observes that Day = weekday, and then is asked to predict the upcoming Barista state (will the barista be Bob or Eve). Then it observes that Sleepiness = sleepy and is asked to predict the upcoming Mood state (will the barista be happy or angry). Finally, it observes that the Weather = sunny and is asked to predict the upcoming Drink state (will the drink be juice or latte). Note that the order of queries is fixed but the order in which situation features are observed is random.

Both observations and queries are represented by one-hot vectors (for details, see the Stimulus representation section in the Materials and methods). In our simulations, the length of the event graph is 16 and the number of states for each time point is 4. This means the number of unique ways in which an event can unfold (depending on the features of the current situation) is $4^{16}$ – far too many to memorize. As such, learning an effective representation of the event graph (i.e., which states can occur at which time points, and how the state transitions depend on the values of the situation features) is essential for predicting which state will come next. In our model, this information is learned during the meta-training phase and stored in the neocortical network's weights (i.e., the model's semantic memory). As a terminological point, in this paper we use the term *situation* to refer to the 'ground

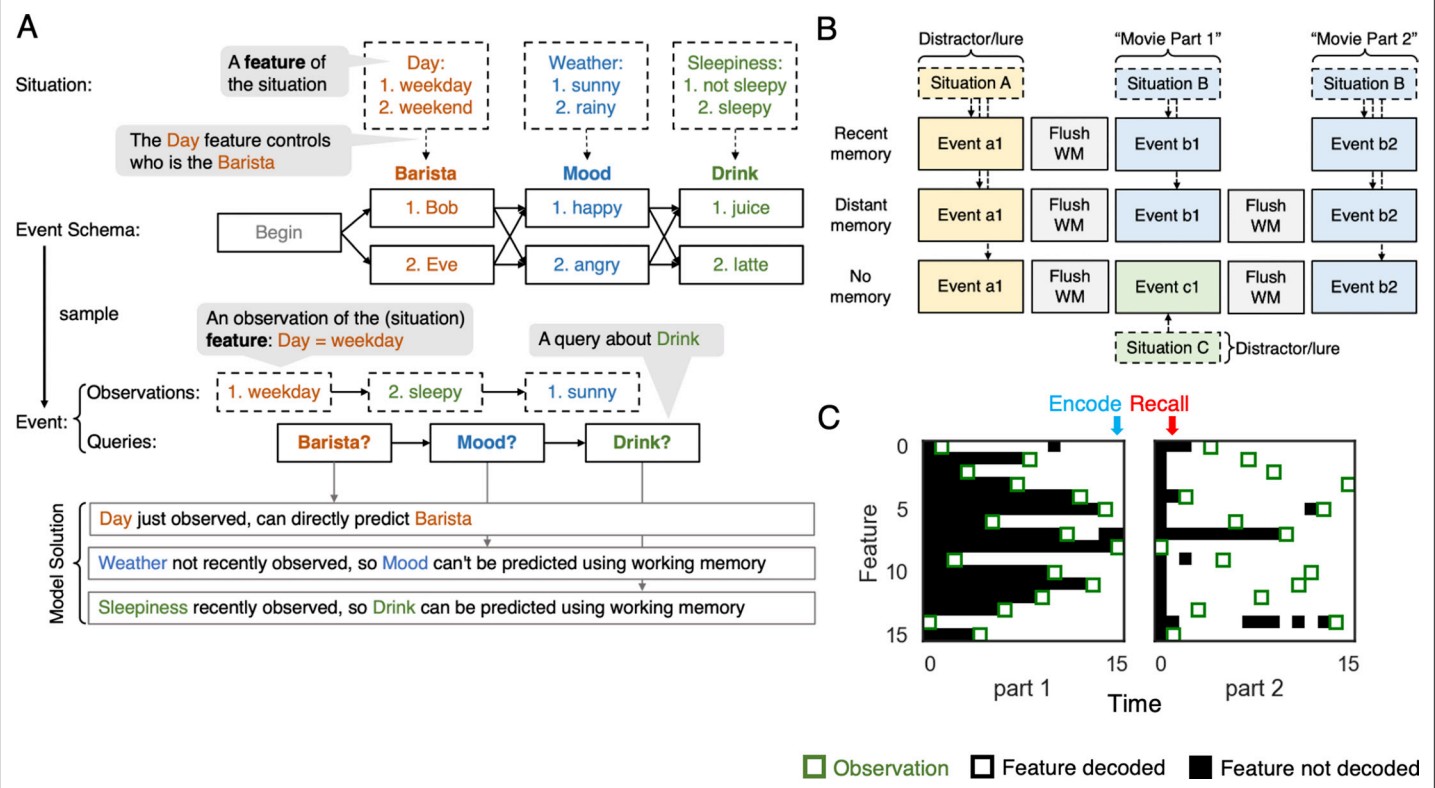

**Figure 2.** A situation-dependent event processing task. (**A**) An *event* is a sequence of states, sampled from an event schema and conditioned on a situation. An *event schema* is a graph where each node is a state. A *situation* is a collection of features (e.g., Day, Weather, Sleepiness) set to particular values (e.g., Day = weekday). The features of the current situation deterministically control how the event unfolds (e.g., the value of the Day feature controls which Barista state is observed). At each time point, the network observes the value of a randomly selected feature of the current situation, and responds to a query about what will happen next. Note that the order of queries is fixed but the order in which situation features are observed is random. If a situation feature (Sleepiness) is observed before the state it controls (Drink), the model can answer the query about that state by holding the relevant feature in working memory. However, if the relevant situation feature (Weather) is observed after the state it controls (Mood), the model can not rely on working memory on its own to answer the query. (**B**) We created three task conditions to simulate the design used by *Chen et al., 2016*: recent memory (RM), distant memory (DM), and no memory (NM); see text for details. (**C**) Decoded contents of the model's working memory for an example trial from the DM condition. Green boxes indicate time points where the value of a particular situation feature was observed. The color of a square indicates whether the correct (i.e., observed) value of that feature can be decoded from the model's working memory state (white = feature accurately decoded; black = feature not decoded). See text for additional explanation.

truth' of the feature-value pairings for the current event, and we use *situation model* to refer to the model's internal representation of the current situation in working memory (i.e., in the LSTM cell state).

*Figure 2A* also illustrates how the memory demands of the task vary depending on when the situation feature that controls a given state is observed, relative to the query for that state. If the situation feature (Day) that controls a particular state (Barista) is observed on the same time step as the query for that state, no memory is needed to answer the query. If the relevant situation feature (Sleepiness) for controlling a state (Drink) was recently observed (e.g., at an earlier time step in the sequence), the model can answer the query by holding the value of that feature in working memory. If the relevant situation feature (Weather) for controlling a state (Mood) was not recently observed, the model can not use working memory on its own to answer the query; here, the only way the model can respond correctly is by retrieving a stored episodic memory of that situation.

*Figure 2B* shows the way we simulated the three conditions from *Chen et al., 2016*. In each of the conditions, the agent processes three events. Importantly, for all of the conditions, we imposed (by hand) an encoding policy where the model stored an episodic memory (reflecting the current contents of working memory – i.e., the LSTM cell state) on the final time point of each event. This encoding policy was based on previous findings suggesting that episodic encoding takes place selectively at the end of an event (*Ben-Yakov and Dudai, 2011*; *Ben-Yakov et al., 2013*; *Baldassano*

*et al., 2017*; *Ben-Yakov and Henson, 2018*; *Reagh et al., 2020*); we critically examine this assumption in the *Benefits of selectively encoding at the end of an event* section below. In both the RM and DM conditions, the agent first processes a distractor event (i.e., event a1), and then processes two related events that are controlled by the same situation (i.e., event b1 and b2). These two related events capture the two-part movie in the study by *Chen et al., 2016*, in the sense that knowing information from the first event (b1) will make the second event (b2) more predictable. Note that, at the start of movie part 2 (b2), models in both the RM and DM conditions have access to a lure episodic memory that was formed during the distractor event (a1), and also a target episodic memory that was formed during movie part 1 (b1). The main difference is that, in the DM condition, the working memory state is flushed between part 1 and part 2 (by resetting the cell state of the LSTM), whereas the flush does not occur in the RM condition; this flush in the DM condition is meant to capture the effects of the one-day delay between part 1 and part 2 in the study by *Chen et al., 2016*. Finally, in the NM condition, the agent processes three events from three different situations. Therefore, during movie part 2, the agent has no information in working memory or episodic memory pertaining to part 1. The model was trained (repeatedly) to predict upcoming states on all three trial types before being tested on each of these trial types (see the *Model training and testing* section in the Materials and methods).

To summarize, the task environment used in our simulations captures how understanding of naturalistic events and narratives depends on memory: It is necessary to remember observations from the past (possibly from a large number of time points ago) in order to optimally predict the future. For example, in the *Twilight Zone* episode used by *Chen et al., 2016*, learning that the servants are robots early in the episode helps the viewer predict how one character will react when another character suggests killing all of the servants; similarly, in the model, learning that the weather is sunny during event b1 will help the model predict that the barista will be happy during event b2. The model is incentivized to routinely hold observations in working memory, because information that is observed early in an event can sometimes be used to answer queries that are posed later in that same event, or possibly across events (in the RM condition). This should lead to a dynamic where the amount of information held in working memory builds within an event (i.e., with each successive observation, the model builds a more 'complete' representation in working memory of the features of the current situation). Episodic memory is incentivized because of the working memory 'flush' in the DM condition between events b1 and b2 – information that is relevant to b2 is observed during b1 but flushed from working memory, so the only way to benefit from this information is to store it in episodic memory (at the end of b1) and then retrieve it from episodic memory at the start of b2 (for additional discussion of how episodic memory can help to bridge interruptions, see classic work by *Ericsson and Kintsch, 1995*).

*Figure 2C* illustrates these points by showing the decoded contents of the model's working memory for an example DM trial. To generate this figure, a linear classifier (logistic regression with L2 penalty) was used to decode whether the correct (i.e., observed) value of each situation feature was represented in the working memory state of the model (i.e., the LSTM cell state) at each time point during the trial; see the *Decoding the working memory state* section in the Materials and methods for more details. We found that, once a feature was observed (indicated by a green box in the figure), this feature typically was decodable until the end of the event, which confirms that observed features tend to be actively maintained in the working memory state of the agent. The figure makes it clear how, because of this tendency to maintain information over time, the model's representation of the situation becomes more complete over time within part 1 of the event. The model then stores an episodic memory snapshot on the final time point in part 1 (indicated by the blue arrow). Between part 1 and part 2, the model's working memory state is flushed; then, early in part 2, the model retrieves the stored episodic memory snapshot (indicated by the red arrow), which results in many features of the situation becoming decodable before they are actually observed during part 2.

We acknowledge that our event-processing simulations incorporate several major simplifications. For example, we are modeling the first part of the movie as a single event when, in the *Chen et al., 2016* study, each half of the Twilight Zone episode clearly contains multiple events. We also are assuming that the rate of key situation features being revealed is linear (one per time point) and that feature values stay stable within events. Our goal here was to come up with the simplest possible framework that allowed us to meaningfully engage with questions about encoding and retrieval

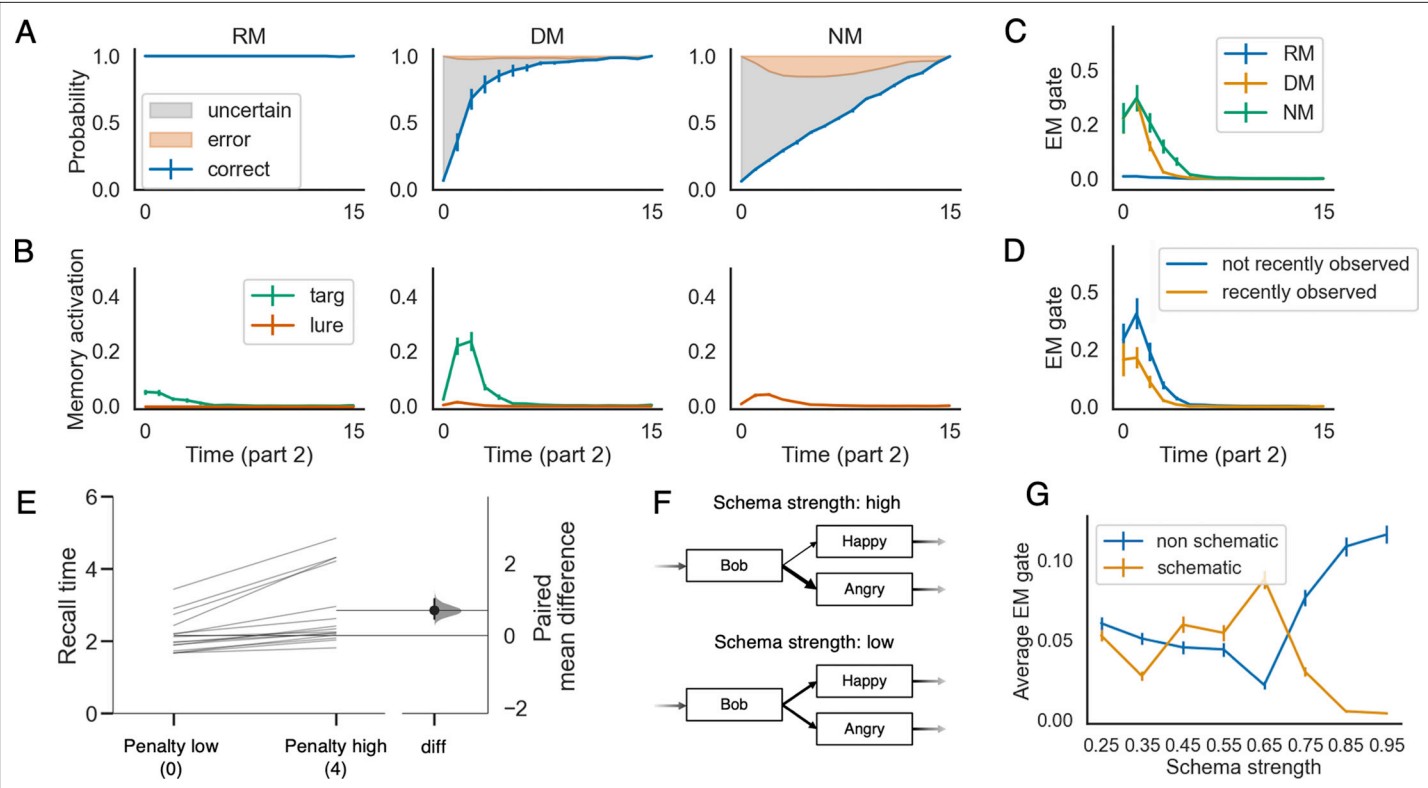

**Figure 3.** The learned episodic retrieval policy is selective. Panels **A, B** and **C** show the model's behavioral performance, memory activation, and episodic memory gate (EM gate) value during part 2, across the recent memory (RM), distant memory (DM), and no memory (NM) conditions, when the penalty for incorrect prediction is set to two at test. These results show that recall is much stronger in the DM condition (where episodic retrieval is needed to fill in gaps and resolve uncertainty) compared to the RM condition. (**D**) shows that, in the DM condition, the EM gate value is lower if the model has recently (i.e., in the current event) observed the feature that controls the upcoming state transition. (**E**) shows how the average recall time is delayed when the penalty for making incorrect predictions is higher. (**F**) illustrates the definition of the schema strength for a given time point. (**G**) shows how the average EM gate value changes as a function of schema strength (penalty level = 2). The errorbars indicate 1SE across 15 models.

policies for episodic memory. In the *Discussion*, we talk about ways that the model could be extended to more fully address the complexity of real-world events.

## The learned retrieval policy is sensitive to uncertainty

*Figure 3A* shows the trained model's prediction performance during movie part 2, with the penalty value for incorrect prediction set to 2. In the recent memory (RM) condition, prediction accuracy is at ceiling starting from the beginning of part 2 – all situation feature values for the ongoing situation were observed during the first part of the sequence, and the model is able to hold on to these features in working memory. In the distant memory (DM) condition, prediction accuracy starts out much lower, but after a few time points the accuracy is almost at ceiling. In the no memory condition (NM), prediction accuracy increases linearly, reflecting the fact that the model is able to observe more situation features as the event unfolds. The fact that prediction accuracy is better in the DM condition than in the NM condition suggests that the model is using episodic memory to support prediction in the DM condition.

We were particularly interested in whether the model's learned retrieval policy would be demand-sensitive (i.e., would the model be more prone to retrieve from episodic memory if there were gaps in its situation model, leading it to be uncertain about the upcoming state). To answer this question, we visualized the activation levels of the target and lure memories during part 2, for each of the three conditions (*Figure 3B*). Across the three conditions, we found much higher levels of memory activation in the DM condition than the other two conditions. Importantly, the finding (in the model) of greater memory activation in the DM condition than the RM condition qualitatively captures the finding from *Chen et al., 2016* that the putative fMRI signature of episodic retrieval (hippocampal-neocortical

coupling) was stronger in the DM condition than the RM condition. Note that, in our simulation, the set of available episodic memories in the RM and the DM condition is the same. The main difference is that, in the RM condition, the network has a fully-specified situation model actively maintained in its working memory (the recurrent activity of the LSTM) during part 2, which is sufficient for the network to predict the upcoming state. In contrast, at the beginning of the DM condition, the network's ongoing situation model is empty – the values for all features are unknown. Overall, this result suggests that the learned retrieval policy is demand-sensitive (for simulations of other, related findings from this study, see *Appendix 6*).

To gain further insight into the model's retrieval policy, we examined the EM gate values in the three conditions (*Figure 3C*). We found that the model sets the EM gate to a higher (more 'open') value in the DM and NM conditions (where there are gaps in the model's understanding of the ongoing situation, causing it to be uncertain about what was coming next), and it suppresses episodic retrieval in the RM condition (where there are no gaps). Likewise, within the DM condition, the model sets the EM gate to a higher value when the feature controlling the next transition has not been recently observed (i.e., the feature is not in working memory, causing the model to be uncertain about what was coming next) vs. if the relevant feature has been recently observed and is therefore active in working memory (*Figure 3D*). The same principle also explains why, for later time points in part 2, the EM gate is set to a lower value in the DM condition than the NM condition (*Figure 3C*) – in the DM condition, episodic retrieval that occurs on earlier time points makes the model more certain on later time points, reducing the demand for episodic retrieval and (consequently) leading to lower EM gate values.

The fact that the model learned a demand-sensitive retrieval policy can be explained in terms of a simple cost-benefit analysis: When the model is unsure about what will happen next, the potential benefits of episodic retrieval are high. In the absence of episodic retrieval, the model will have to guess or say 'don't know', but if it consults episodic memory, the model could end up recalling the feature of the situation that controls the upcoming state transition, allowing it to make a correct prediction. By contrast, when the feature of the situation that controls the transition is already in working memory (and consequently the model is able to make a specific prediction about what will happen next), there is less of a benefit associated with episodic retrieval – the only way that episodic retrieval will help is if the model is holding the wrong feature in working memory and the episodic memory overwrites it. Furthermore, in this scenario, there is also a potential cost to retrieving from episodic memory: Lures are always present, and if the model recalls a lure this can overwrite the correct information in working memory. Since the potential costs of episodic retrieval outweigh the benefits of episodic retrieval in the 'high certainty' scenario, the model learns a policy of waiting to retrieve until it is uncertain about what will happen next.

Importantly, the model's ability to *adjust its policy* when it is uncertain is predicated on there being a reliable 'neural correlate of certainty' in the model, which can be used as the basis for this differential responding; we investigated this and found that the norm of activity in the decision layer is lower when the model is uncertain vs. certain (for more details, see *Appendix 1*). This (implicit) neural correlate of certainty exists regardless of whether the model is trained to explicitly signal uncertainty via the 'don't know' response. In other simulations (reported in *Appendix 5*), we found that a version of the model without the 'don't know' option can still leverage this implicit neural correlate of certainty to show demand-sensitive retrieval (i.e., more episodic retrieval in the DM condition than the RM condition); the main effect of including the 'don't know' option is to make the model more patient overall, by reducing the cost associated with waiting to retrieve from episodic memory.

## The effect of penalty on retrieval policy

A key question is how the model's policy for prediction and episodic retrieval adapts to different environmental regimes. Toward this end, we explored what happens when we vary the penalty on false recall from 0 to 4 during model meta-testing – that is, can the model flexibly adjust its policy based on the current penalty? (note that the penalty was uniformly sampled from the 0–4 range during meta-training). If learning a selective retrieval policy is driven by the need to manage the costs of false recall, then it stands to reason that varying these costs should affect the model's policy. Our first finding is that adjusting the penalty at test affects the model's tendency to give 'don't know' responses: When the penalty is zero, the model makes specific next-state predictions (i.e., it refrains from using the 'don't know' response) even when it can not reliably predict the next state, leading to many errors. In

contrast, when the penalty is high, the model makes more 'don't know' responses (in the DM condition, the model responds 'don't know' 15.8% of the time when penalty is set to 4, vs 0.3% of the time when penalty is set to 0). This strategy is rational – when the penalty is zero, the expected reward is larger for randomly guessing an answer than for saying 'don't know', but when the penalty is set to four, the expected reward is larger for saying 'don't know' than for random guessing. We also found that, when the model is tested in an environment where the penalty is high, it waits longer to retrieve from episodic memory, relative to when the penalty at training is lower (*Figure 3E*). This delay in recall can be explained in terms of a speed-accuracy trade-off. Waiting longer to retrieve from episodic memory allows the model to observe more features, which helps to disambiguate the present situation from other, related situations and thereby reduces false recall. However, waiting longer also carries an opportunity cost – the model has to forego all of the rewards it would have received (from correct prediction) if it had recalled the correct memory earlier. When the penalty is low, the benefits of retrieving early (in terms of increased correct prediction) outweigh the costs (in terms of increased incorrect prediction due to false recall), but when the penalty is high, the costs outweigh the benefits, so the model is more cautious and it waits to observe more features to be sure that the memory it (eventually) recalls is the right one.

## The effect of schema regularity on the learned policy

Next, we examined the effect of schema regularity on the agent's retrieval policy. In the simulations preceding this one, we imposed a form of schematic structure by teaching the model about which states could be visited at which time points (i.e., the 'columns' of *Figure 2A*). However, *within* a particular time point, the marginal probabilities of the states that were 'allowed' at that time point were equated – put another way, none of the states were more prototypical than any of the other states. In this simulation, we also allowed for some states to be more prototypical (i.e., occur more often) than other states that could occur at that time point. We say that a time point is *schematic* if there is one state that happens with higher probability, compared to other states. Consider the example illustrated in *Figure 3F*: If the probability of Bob being angry is much greater than the probability of him being happy, then we say that this is a highly schematic time point. In contrast, if Bob is equally likely to be happy or angry, then the schema strength is low. Intuitively, when there is a strong schema, there is less of a need to rely on episodic memory – in the limiting case, if the schematic state occurs in every sequence, the model will learn to predict this state every time and there is no need to consult episodic memory.

To explore the effects of schema strength, we ran simulations where half of the time points were schematic. For the other half of the time points (*non-schematic* time points), all of the states associated with that time point were equally probable (given that there were four possible states at each time point, the probability of each state was 0.25). Schematic and non-schematic time points were arranged in an alternating fashion (for half of the models, even time points were schematic and odd time points were non-schematic, and the opposite was true for the other half of the models). For schematic time points, we manipulated the strength of schematic regularity in the environment by manipulating the probability of the 'prototypical' state. We varied schema strength values from 0.25 (baseline) to 0.95 in steps of 0.10.

The results of this analysis when penalty was set to two at test are shown in *Figure 3G*, which plots the EM gate value during part 2 as a function of schema strength. The first thing to note about these results is that, for high levels of schema strength, episodic retrieval is suppressed for schematic time points (i.e., time points with a prototypical state) and elevated for non-schematic time points (i.e., time points where there was not a prototypical state). The former finding (suppression of retrieval at time points where there is a strong prototype) fits with the intuition, noted above, that high-schema-strength states are almost fully predictable without episodic memory, and thus there is no need to retrieve from episodic memory. The latter finding (enhanced retrieval at non-schematic time points, when schema strength is high overall) can be explained in terms of the idea that schema-congruent features tend to be shared by both target and lure memories and thus are not diagnostic of which memory is the target; in this situation, the only way to distinguish between targets and lures is to recall non-schematic features, which is why the model tries extra-hard to retrieve them from episodic memory.

Interestingly, the model shows the opposite pattern of effects when schema strength = .55 or .65: Episodic retrieval is enhanced for schematic time points and suppressed for non-schematic time points. This reversal can be explained as follows: When schema strength = .55 or .65, the model has started to build up a tendency to guess the schema-congruent (prototypical) state, but it is also going to be wrong about 1/3 of the time when it guesses the schema-congruent state, incurring a substantial penalty. To counteract this tendency to make wrong guesses, the model needs to try extra-hard to retrieve the actual feature value for schematic time points (which is why the EM gate value increases for these time points) – and if the model is doing more retrieval in response to schematic states, it needs to do somewhat less retrieval in response to non-schematic states (which is why the EM gate value goes down for these features). As schema strength increases beyond .65, the model will be wrong less often when it guesses the schema-congruent state, so there is less of a need to counteract wrong guesses with episodic retrieval – this makes it safe for the model to reduce the EM gate value for schematic time points at higher levels of schema strength (as described above).

## Other factors that affect the learned retrieval policy

In addition to the simulations described above, we also ran simulations exploring the effects of *between-event similarity* and *familiarity* on the learned retrieval policy. With regard to similarity: We found that the model is more cautious about retrieving from episodic memory if trained in environments where memories are highly similar (because the risk of false recall is higher) – see *Appendix 2* for details. With regard to familiarity: When we provided the model with a familiarity signal that is informative about whether a situation was previously encountered, we found that the model learns to exploit this information by retrieving more from episodic memory when the familiarity signal is high and retrieving less from episodic memory when the familiarity signal is low. This result provides a resource-rational account of experimental findings showing that familiar stimuli shift the hippocampus into a 'retrieval mode' where it is more likely to (subsequently) retrieve episodic memories (*Duncan et al., 2012*; *Duncan and Shohamy, 2016*; *Duncan et al., 2019*; *Patil and Duncan, 2018*; *Hasselmo and Wyble, 1997*; *Hasselmo et al., 1995*) – see *Appendix 3* for details.

## Benefits of selective encoding

Above, we showed that the model learned selective retrieval policies (e.g., avoiding retrieval from episodic memory early on during part 2, or when certain about upcoming states) in order to reduce the risk of recalling irrelevant memories. Here, we shift our focus to the complementary question of *encoding policy*: When is the best time to store episodic memories? In the simulations reported below, we show that a selective encoding policy can benefit performance, by reducing interference at retrieval later on. Note that our model is presently not capable of learning an encoding policy on its own (see *Discussion*), but we can explore the benefits of selective encoding by imposing different encoding policies by hand and seeing how they affect performance.

### Benefits of selectively encoding at the end of an event

The simulations presented thus far assumed that episodic memories are selectively encoded at the ends of events. This assumption was based on findings from several recent fMRI studies that measured hippocampal activity during perception of events and related this to later memory for the events. These studies found that the hippocampal response tends to peak at event boundaries (*Ben-Yakov and Dudai, 2011*; *Ben-Yakov et al., 2013*; *Baldassano et al., 2017*; *Ben-Yakov and Henson, 2018*; *Reagh et al., 2020*); this boundary-locked response predicts subsequent memory performance for the just-completed event (*Ben-Yakov and Dudai, 2011*; *Baldassano et al., 2017*; *Reagh et al., 2020*), leading researchers to conclude that it is a neural signature of episodic encoding of the just-completed event.

While these results suggest that the end of an event may be a particularly important time for episodic encoding, existing studies do not provide a computational account of *why* this should be the case. This 'why' question can be broken into two parts: First, why might it be beneficial to encode at the end of an event, and second, why might it be *harmful* to encode at other times within the event? Answering the first question (regarding benefits of encoding at the end of an event) is relatively straightforward. Several researchers have argued that information about the current situation builds up in working memory within an event, and then is 'flushed out' at event boundaries (*Radvansky*

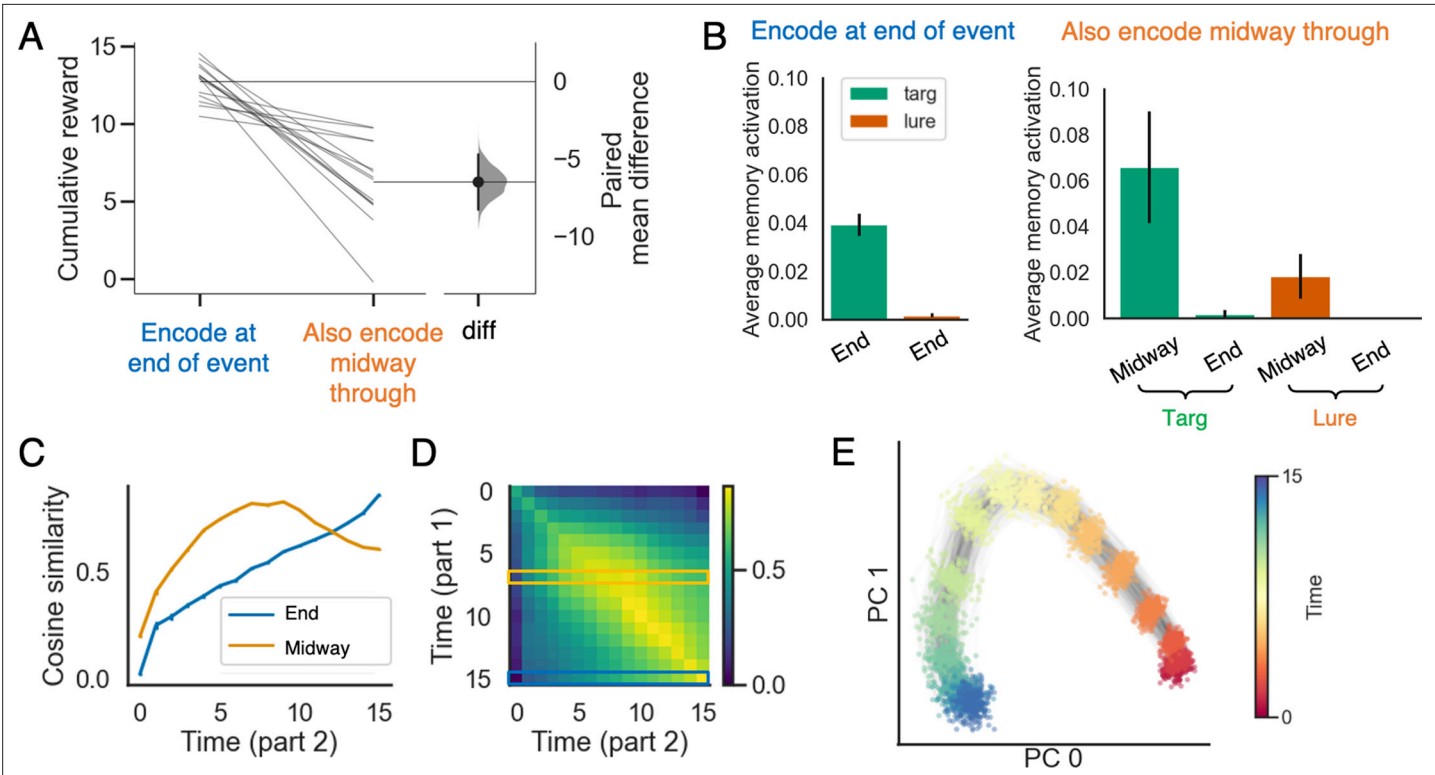

**Figure 4.** The advantage of selectively encoding episodic memories at the end of an event. (**A**) Prediction performance is better for models that selectively encode at the end of each event, compared to models that encode at the end of each event and also midway through each event. (**B**) The model performs worse with midway-encoded memories because midway-encoded target memories are activated more strongly than end-encoded target memories, thereby blocking recall of the (more informative) end-encoded target memories, and also because midway-encoded lure memories are more strongly activated than end-encoded lure memories (see text for additional discussion). (**C**) The cosine similarity between working memory states during part 2 and memories formed midway through part 1 (in orange) or at the end of part 1 (in blue). The result indicates that the midway-encoded memory will dominate the end-encoded memory for most time points. (**D**) The time-point-to-time-point cosine similarity matrix between working memory states from part 1 versus part 2 in the no memory (NM) condition (part C depicts the orange and blue rows from this matrix). (**E**) PCA plot of working memory states as a function of time, for a large number of events. The plot shows that differences in time within events are represented much more strongly than differences across events. The errorbars indicate 1SE across 15 models.

et al., 2011; *Richmond and Zacks, 2017*; for neural evidence in support of this dynamic, see *Ezzyat and Davachi, 2011*; *Chien and Honey, 2020*; *Ezzyat and Davachi, 2021*). This dynamic (which is illustrated in the model in *Figure 2C*) means that the model's representation of the features of an event will be most complete right before the end of the event, making this a particularly advantageous time to take an episodic memory snapshot of the situation model.

While it is clear why encoding at the end of an event is useful, it is less clear why encoding at other times might be harmful; naively, one might think that storing more episodic snapshots during an event would lead to *better* memory for the event. To answer this question, we compared models that selectively encode episodic memories at the end of each event to models that encode episodic memories both at the end of each event and also midway through each event. Note that the model did not learn these two encoding policies by itself – we simply configured the model to encode one way or the other and compared their performance. If selectively encoding at the end of an event yields better performance, this would provide a resource-rational justification for the empirical findings reviewed above.

Our simulation shows that in the DM condition, during part 2, models that encode an additional episodic memory midway through each event performed worse (*Figure 4A*). This decrease in performance can be explained in terms of several related factors. First, as shown in *Figure 4B*, when midway memories are also stored, midway memories of the target event are recalled more strongly than memories formed at the end of the target event.

This advantage occurs because the model's hidden state strongly encodes temporal context: WM states stored at similar times within an event tend to be more similar than WM states stored at different times (this illustrated by *Figure 4E*, which shows that time information within an event is more strongly represented than differences across events). This strong temporal coding makes sense, given that the model needs to know where it is in the sequence in order to predict which observations will come next (for evidence for this kind of temporal coding in the brain, see *Pastalkova et al., 2008*; *MacDonald et al., 2011*; *Salz et al., 2016*; *Tiganj et al., 2017*; for reviews, see *Eichenbaum, 2014*; *Howard and Eichenbaum, 2013*; for models of this temporal coding, see *Shankar and Howard, 2012*; *Howard et al., 2014*; *Liu et al., 2019*). One consequence of this time coding is that – early on in part 2 of the event (when the benefits of episodic retrieval are the largest) – the temporal context represented in working memory will be a better match for memories encoded midway through the event than memories encoded at the end of the event (*Figure 4C and D*). This temporal context match provides a competitive advantage for the midway memory over the endpoint memory, resulting in the midway memory blocking the endpoint memory from coming strongly to mind. The second key point is that the midway memory is less informative (i.e., it contains fewer features of the situation, because it was stored before the full set of features was observed). As such, recalling the midway target memory confers less of a benefit on future prediction than recalling the endpoint memory would have provided – this is the main reason why prediction is worse in the midway condition. The third key point is that, because midway memories contain less information, they are more confusable across events (i.e., it is harder to determine which event the memory pertains to). As a result, midway lures tend to become more active at retrieval than endpoint lures (*Figure 4B*) – this lure retrieval acts to further reduce prediction accuracy.

A possible alternative explanation of the negative effects of midway encoding is that midway encoding was introduced when we tested the model's performance but was not present during meta-training (i.e., when the model acquired its retrieval policy); as such, midway encoding can be viewed as 'out of distribution' and may be harmful for this reason. To address this concern, we also ran a version of the model where memories were stored both midway and at the end of an event during meta-training, and it was still true that endpoint-only encoding led to better performance than midway-plus-endpoint encoding; this result shows that midway encoding is intrinsically harmful, and it is not just a matter of it being out-of-distribution. In another simulation, we also found that the harmful effect of encoding midway through the sequence qualitatively replicates with more complex event structures (analogous to those in *Botvinick and Plaut, 2004*) where queries are repeated within a sequence and the model has to give a different response to the first vs. second occurrence of a query (e.g., mood is controlled first by weather and later by music).

To summarize the results from this section, the model does better when we force it to wait until the end of an event to take a snapshot. This pattern arises in our simulations because knowledge of the ongoing situation builds within an event, so memories encoded earlier in the event contain less information about the situation than memories encoded at the end. These less-informative midway memories harm performance by interfering with recall of more-informative endpoint memories, and are themselves more prone to false recall (because they are more confusable across events). Taken together, these simulation results provide a resource-rational justification for the results cited above showing preferential encoding-related hippocampal activity at the end of events (*Ben-Yakov and Dudai, 2011*; *Ben-Yakov et al., 2013*; *Baldassano et al., 2017*; *Ben-Yakov and Henson, 2018*; *Reagh et al., 2020*).

## Discussion

Most of what we know about episodic memory has, by design, come from experiments where performance depends primarily on episodic memory (as opposed to other memory systems), and participants are given clear instructions about when episodic memories should be stored and retrieved (e.g., learning and recalling lists of random word pairs); likewise, most computational models of human memory have focused on explaining findings from these kinds of experiments (e.g., see *Gillund and Shiffrin, 1984*; *Hasselmo and Wyble, 1997*; *Shiffrin and Steyvers, 1997*; *Howard and Kahana, 2002*; *Norman and O'Reilly, 2003*; *Sederberg et al., 2008*; *Polyn et al., 2009*; *Cox and Criss, 2020*; for reviews, see *Norman et al., 2008*; *Criss and Howard, 2015*). However, as noted in the *Introduction*, real-world memory does not adhere to these constraints: In naturalistic learning situations,

participants are typically not given any instructions about how episodic memory should be used to support performance, and – even when participants are given instructions about what to remember – performance usually depends on a complex mix of memory systems, with contributions from both working memory and semantic memory in addition to episodic memory.

The goal of the present work was to gain some theoretical traction on when episodic memories should be stored and retrieved to optimize performance in these more complex situations. Towards this end, we optimized a neural network model that *learned its own policy* for when to consult episodic memory (via an adjustable gate) in order to maximize reward, and we also (by hand) explored the effects of different episodic memory encoding policies on network performance. Our approach is built on the principle of resource rationality, whereby human cognition is viewed as an approximately optimal solution to the learning challenges posed by the environment, subject to constraints imposed by our cognitive architecture (*Griffiths et al., 2015*; *Lieder and Griffiths, 2019*); according to this principle, the approximately optimal solutions obtained by our model can be viewed as hypotheses about (and explanations of) how humans use episodic memory in complex, real-world tasks.

In the simulations presented here, we identified several ways in which selective policies for episodic memory retrieval and encoding can benefit performance. With regard to retrieval, we showed that the model learns to avoid episodic retrieval in situations where the risks of retrieval (i.e., retrieving the wrong memory, leading to incorrect predictions) outweigh the benefits (i.e., retrieving the correct memory, leading to increased correct predictions). For example, when there is high certainty about what will be observed next (due to the relevant information being maintained in working memory or semantic memory), the marginal benefits of retrieving from episodic memory are too small to outweigh the risks of retrieving the wrong memory. Another example is when too little information has been observed to pinpoint the relevant memory – in this case, the potential benefits of retrieving are high, but the risks of retrieving the wrong memory are also high, leading the model to defer retrieving until more information has been observed. With regard to encoding, we showed that waiting until the end of an event to encode a memory for that event boosts subsequent prediction performance – this performance boost comes from reducing 'clutter' (interference) from other memories, thereby making it easier to retrieve the sought-after memory. These modeling results explain a wide range of existing behavioral and neuroimaging results, and also lead to new, testable predictions. With regard to existing results: The model provides a resource-rational account of findings from *Chen et al., 2016* showing the demand-sensitivity of episodic retrieval, as well as results showing that episodic encoding is modulated by event boundaries (*Ben-Yakov and Dudai, 2011*; *Ben-Yakov et al., 2013*; *Baldassano et al., 2017*; *Ben-Yakov and Henson, 2018*; *Reagh et al., 2020*). *Appendix 3* also shows how the model explains effects of familiarity on retrieval policy (*Duncan et al., 2012*; *Duncan and Shohamy, 2016*; *Duncan et al., 2019*; *Patil and Duncan, 2018*; *Hasselmo and Wyble, 1997*). With regard to novel predictions: Our model makes predictions about how episodic retrieval will be modulated by certainty (*Figure 3B, C and D*), penalty (*Figure 3E*), schema strength (*Figure 3G*), and similarity (*Appendix 2—figure 1*) – all of these predicted relationships could be tested in experiments that measure hippocampal-neocortical information transfer, either using measures like hippocampal-neocortical inter-subject functional connectivity in fMRI (e.g., *Chen et al., 2016*; *Chang et al., 2021*) or time-lagged mutual information in ECoG (e.g., *Michelmann et al., 2021*).

More broadly, the simulations presented here show how the model can be used to explore interactions between three distinct memory systems: semantic memory (instantiated in the weights in neocortex), working memory (instantiated in the gating policy learned by the neocortical LSTM module, allowing for activation at one time point in neocortex to influence activation at subsequent time points), and episodic memory. In the past, modelers have focused on these memory systems in isolation (see, e.g., *Norman et al., 2008*), in part because of a desire to understand the detailed workings of the systems, but also because of technical limitations: Until very recently, the technology did not exist to automatically optimize the performance of networks containing episodic memory, so researchers interested in simulating interactions between episodic memory and these other systems were put in the position of having to do time-consuming (and frustrating) hand-optimization of the models. Here, we leverage recent progress in the artificial intelligence literature on memory-augmented neural networks (*Graves et al., 2016*; *Pritzel et al., 2017*; *Ritter et al., 2018*; *Wayne et al., 2018*) that makes it possible to automatically optimize the use of episodic memory and its interactions with other memory systems. This technical advance has opened up a new frontier in

the cognitive modeling of memory (*Collins, 2019*), making it possible to address both 'naturalistic memory' scenarios and controlled experiments that involve interactions between prior knowledge (semantic memory), active maintenance (working memory), and episodic memory.

## Relation to other models

### Memory-augmented neural networks with a differentiable neural dictionary

Conceptually, the episodic memory system used in our model is similar to recently-described memory-augmented neural networks with a differentiable neural dictionary (DND) (*Pritzel et al., 2017*; *Ritter et al., 2018*; *Ritter, 2019*). In these models, the data structure of the episodic memory system is dictionary-like: Each memory is a key-value pair. The keys define the similarity metric across all memories, and the values represent the content of these memories. For example, one can use the LSTM cell state patterns as the keys and use the final output of the network as the values (*Pritzel et al., 2017*). *Ritter et al., 2018* is particularly relevant as it was the first paper (to our knowledge) to use the DND for cognitive modeling and – as such – served as a major inspiration for the work presented here (see also *Botvinick et al., 2019*). The way that our model uses the DND mechanism is quite similar to how it was used in *Ritter et al., 2018*; in particular, we took from the *Ritter et al., 2018* paper the idea that the neocortical network learns to control a 'gate' on episodic retrieval via reinforcement learning (for an earlier model that also used reinforcement learning to learn a policy for retrieval from episodic memory, see *Zilli and Hasselmo, 2008*). However, there are also some meaningful differences between our model and the model used by *Ritter et al., 2018*.

The most salient difference regards the placement of the EM gate: In our model, the gate controls the flow of information into the episodic memory module (*pre-gating*), but in the Ritter model the gate controls the flow of information *out* of the episodic memory module (*post-gating*). Practically speaking, the main consequence of having the gate on the output side is that the gate can be controlled based on information coming out of the hippocampus, in addition to all of the neocortical regions that are used to control the gate in our pre-gating model. While this is a major difference, we found that our key simulation results qualitatively replicate in a version of the model that uses post-gating, indicating that the selective encoding and retrieval principles discussed here do not depend on the exact placement of the gate (see *Appendix 4* for simulation results and more discussion of these points).

Another difference is that our model's computation of which memories are retrieved (given a particular retrieval cue, assuming that the 'gate' on retrieval is open) is more complex. *Ritter et al., 2018* used a one-nearest-neighbor matching algorithm during recall, whereby the stored memory with the highest match to the cue is selected for retrieval (assuming that the gate is open). By contrast, memory activation in our model is computed using a competitive evidence accumulation process, in line with prior cognitive models of retrieval (e.g., *Polyn et al., 2009*; *Sederberg et al., 2008*). While we did not explore the effects of varying the level of competition in our simulations, having this as an adjustable parameter opens the door to future work where the model learns a policy for setting competition in order to optimize performance (just as it presently learns a policy for setting the EM gate).

A third structural difference between our model and the *Ritter et al., 2018* model is our addition of the 'don't know' output unit, which (when selected) allows the model to avoid both reward and punishment. As discussed above, the primary effect of incorporating this 'don't know' action is to make the model more patient (i.e., more likely to wait to retrieve from episodic memory), by giving it a way to avoid incurring penalties if it decides to wait to retrieve (for more details, see *Appendix 5*).

Apart from the structural differences noted above, the main difference between our modeling work and the work done by *Ritter et al., 2018* relates to the application domain (i.e., which cognitive phenomena were simulated). Our modeling work in this paper focused on how episodic memory can support incidental prediction of upcoming states, when there is no explicit demand for a decision. By contrast, *Ritter et al., 2018* focused on how episodic memory can be used to support performance in classic decision-making tasks, such as bandit tasks and maze learning, that have been extensively explored in the reinforcement learning literature.

### The structured event memory (SEM) model

Another highly relevant model is the structured event memory (SEM) model developed by *Franklin et al., 2020*. Like our model, SEM uses RNNs to represent its knowledge of schemas (i.e., how events

typically unfold). Also, like our model, SEM records episodic memory traces as it processes events. However, there are several key differences between our model and SEM. First, whereas our model uses a single RNN to represent a single (contextually parameterized) schema, SEM uses multiple RNNs that each represent a distinct schema for how events can unfold. Building on prior work on nonparametric Bayesian inference (*Anderson, 1991*; *Aldous, 1983*; *Pitman, 2006*) and latent cause modeling (*Gershman et al., 2010*; *Gershman et al., 2015*), SEM contains specialized computational machinery that allows it to determine which of its stored schemas (each with its own RNN) is relevant at a particular moment, and also when it is appropriate to instantiate a new schema (with its own, new RNN) to learn about ongoing events. This inference machinery allows SEM to infer when event boundaries (i.e., switches in the relevant schema) have occurred; the *Franklin et al., 2020* paper leverages this to account for data on how people segment events. Our model lacks this inference machinery, so we need to impose event boundaries by fiat, as opposed to having the model identify them on their own.

Another major difference between the models relates to how episodic memory is used. A key focus of our modeling work in this paper is on how episodic memory can support online prediction. By contrast, in SEM, episodic memory is not used at all for online prediction – online prediction is based purely on the weights of the RNNs (i.e., semantic memory) and the activation patterns in the RNNs (i.e., working memory). The sole use of episodic memory in the *Franklin et al., 2020* paper is to support reconstruction of previously-experienced events. Specifically, in SEM, each time point leaves behind a noisy episodic trace; the *Franklin et al., 2020* paper shows how Bayesian inference can combine these noisy stored episodic memory traces with stored knowledge about how events typically unfold (in the RNNs) to reconstruct an event. Effectively, SEM uses knowledge in the RNNs to 'de-noise' and fill in gaps in the stored episodic traces. The *Franklin et al., 2020* paper uses this process to account for several findings relating to human reconstructive memory.

## Future directions and limitations

On the modeling side, our work can be extended in several different ways. As noted above, our model and SEM have complementary strengths: SEM is capable of storing multiple schemas and doing event segmentation, whereas our model only stores a single schema and we impose event boundaries by hand; our model is capable of using episodic memory to support online prediction, whereas SEM is not. It is easy to see how these complementary strengths could be combined into a single model: By adding SEM's ability to do multi-schema inference to our model, we would be able to simulate both event segmentation and the role of episodic memory in predicting upcoming states, and we would also be able to explore *interactions* between these processes (e.g., using episodic memory to predict could affect when prediction errors occur, which – in turn – could affect how events are segmented; *Zacks et al., 2011*; *Zacks et al., 2007*).

Another limitation of the current model is that the encoding policy is not learned. In our simulations, we trained models with different (pre-specified) encoding policies and compared their performance. Going forward, we would like to develop models that learn when to encode through experience, instead of imposing encoding policies by hand. Our results show that selective encoding can yield better performance than encoding everything, so – in principle – selective encoding policies should be learnable with RL. The main challenge in learning encoding policies is the long temporal gap between the decision to encode (or not) and learning the consequences of that choice for retrieval. Moreover, a high-quality encoding policy, taken on its own, generally does not lead to high reward when the retrieval policy is bad; that is, encoding policy and retrieval policy have to be learned in a highly coordinated fashion. Recent technical advances in RL (e.g., algorithms that do credit assignment across long temporal gaps; *Raposo et al., 2021*) may make it easier to address these challenges going forward.

A benefit of being able to learn encoding policies in response to different task demands is that the model could discover other factors that it could use to modulate encoding – for example, surprise. Numerous studies have found improved memory for surprising events (e.g., *Greve et al., 2017*; *Greve et al., 2019*; *Quent et al., 2021a*; *Kafkas and Montaldi, 2018*; *Frank et al., 2020*; *Rouhani et al., 2018*; *Rouhani et al., 2020*; *Chen et al., 2015a*; *Pine et al., 2018*; *Antony et al., 2021*; for reviews, see *Frank and Kafkas, 2021a*; *Quent et al., 2021b*) – these behavioral results converge with a large body of literature showing increased hippocampal engagement in response to prediction error

(e.g., *Axmacher et al., 2010*; *Chen et al., 2015a*; *Long et al., 2016*; *Kumaran and Maguire, 2006*; *Kumaran and Maguire, 2007*; *Duncan et al., 2012*; *Davidow et al., 2016*; *Kafkas and Montaldi, 2015*; *Frank et al., 2021b*; for reviews, see *Frank and Kafkas, 2021a*; *Quent et al., 2021b*), and also with a recent fMRI study showing that prediction error biases hippocampal dynamics towards encoding (*Bein et al., 2020*). Given that studies have found a strong relationship between surprise and event segmentation (e.g., *Zacks et al., 2011*; *Zacks et al., 2007*; for a recent example see *Antony et al., 2021*), it seems possible that increased episodic encoding at the ends of events could be driven by peaks in surprise that occur at event boundaries. However, there are complications to this view; in particular, some recent work has argued that not all event boundaries are surprising (*Schapiro et al., 2013*) – in light of this, more research is needed to explore the relationship between these effects.

In addition to surprise, recent work by *Sherman and Turk-Browne, 2020* suggests that *predictive certainty* may play a role in shaping encoding policy: They found that stimuli that trigger strong predictions (i.e., high certainty about upcoming events) are encoded less well. In keeping with this point, *Bonasia et al., 2018* found that, during episodic encoding, events that were more typical (and thus were associated with more predictive certainty, and less surprise) were associated with lower levels of medial temporal lobe (MTL) activation. Intuitively, it makes sense to focus episodic encoding on time periods where there is high surprise and low predictive certainty – if events in a sequence are unsurprising and associated with high predictive certainty, this means that existing (neocortical) schemas are sufficient to reconstruct that event, and no new learning is necessary (or, if learning is required, it is possible that neocortex could handle this 'schema-consistent' learning on its own; *McClelland, 2013*; *McClelland et al., 2020*). Conversely, if events in a sequence do not follow a schema (leading to uncertainty) or violate that schema (leading to surprise), the only way to predict those events later will be to store them in episodic memory. Future work can explore whether a model that represents surprise and certainty (either implicitly or explicitly) can learn to leverage one or both of these factors when deciding when to encode; our present model is a good place to start in this regard, as we have already demonstrated the model's ability to factor certainty into its retrieval policy.

Another major simplification in the model's encoding policy is that it stores each episodic memory as a distinct entity (see *Figure 1B*). Old memories are never overwritten or updated. However, a growing literature on memory reconsolidation suggests that memory reminders can result in participants accessing an existing memory and then updating that memory, rather than forming a new memory outright (*Dudai and Eisenberg, 2004*; *Dudai, 2009*; *Hardt et al., 2010*; *Wang and Morris, 2010*). In the future, we would like to develop models that decide whether to encode a new episodic memory or update an old memory. We could implement this by having the model try to retrieve before it encodes a new memory; if it succeeds in retrieving a stored memory above a certain threshold level of activation, the model could update that memory rather than creating a new memory. In future work, we plan to implement this mechanism and use it to simulate memory reconsolidation data.

Going forward, we also hope to explore more biologically-realistic episodic memory models (e.g., *Hasselmo and Wyble, 1997*; *Schapiro et al., 2017*; *Norman and O'Reilly, 2003*; *Ketz et al., 2013*). Using a more biologically realistic hippocampus could affect the model's predictions (e.g., if memory traces were allowed to interfere with each other during storage – currently they only interfere at retrieval) and it would also improve our ability to connect the model to neural data on hippocampal codes and how they change with learning (e.g., *Duncan and Schlichting, 2018*; *Brunec et al., 2020*; *Ritvo et al., 2019*; *Favila et al., 2016*; *Chanales et al., 2017*; *Schlichting et al., 2015*; *Whittington et al., 2020*; *Stachenfeld et al., 2017*; *Hulbert and Norman, 2015*; *Kim et al., 2017*; *Schapiro et al., 2012*; *Schapiro et al., 2016*). Similarly, using a more biologically-detailed neocortical model (separated into distinct neocortical sub-regions) could help us to connect to data on how different neocortical regions interact with hippocampus during event processing (e.g., *Ranganath and Ritchey, 2012*; *Cooper and Ritchey, 2020*; *Ritchey and Cooper, 2020*; *Barnett et al., 2020*; *Gilboa and Marlatte, 2017*; *van Kesteren et al., 2012*; *Preston and Eichenbaum, 2013*). More generally, the simplified nature of the present model (e.g., using rate-coded neurons, training with gradient descent) limits its ability to connect to circuit-level data. Including more detailed circuit-level mechanisms would allow us to leverage the wealth of data that exist at this level to constrain the model. For example, work by Hasselmo and colleagues has shown how cholinergic projections from the basal forebrain to the hippocampus can shift hippocampal dynamics between encoding and retrieval based on whether the retrieval cue is

novel or familiar (*Hasselmo et al., 1995*; *Hasselmo et al., 1996*; *Hasselmo and Wyble, 1997*); this work could be used to inform our simulations (discussed in *Appendix 3*) of how familiarity signals modulate hippocampal retrieval policy. We have opted to start with the simplified episodic memory system described in this paper both for reasons of scientific parsimony and also for practical reasons – adding additional neurobiological details would make the model run too slowly (the current model takes on the order of hours to run on standard computers; adding more complexity would shift this to days or weeks).

Just as our model contains some key simplifications, the environment used in the event processing task is relatively simple and do not capture the full richness of naturalistic events. Some recent studies have explored event graphs with more realistic structure (e.g., *Elman and McRae, 2019*). The fact that our model can presently only handle one schema substantially limits the complexity of the sequences it can process; adding the ability to handle multiple schemas (as discussed above) will help to address this limitation. Also, natural events unfold over multiple timescales. For example, going to the parking lot is an event that involves finding the key, getting to the elevator, etc., but this can be viewed as part of a higher-level event, such as going to an airport. In our simulation, events only have one timescale. In general, introducing additional hierarchical structure to the stimuli would enrich the task demands and lead to interesting modeling challenges. For now, we have avoided more complex task environments for computational tractability reasons, but – as computational resources continue to grow – we hope to be able to investigate richer and more realistic task environments going forward. At the same time, we also plan to use the model to address selective retrieval and encoding effects in list-learning studies (e.g., the aforementioned studies showing that surprise boosts encoding; for reviews, see *Frank and Kafkas, 2021a*; *Quent et al., 2021b*).

Another limitation of the model is that the policies explored here (having to do with when episodic memory snapshots are stored and retrieved) do not encompass the full range of ways in which the use of episodic memory can be optimized. For example, in addition to questions about *when* to encode and retrieve, one can consider optimizations of what is stored in memory and how memory is cued. These kinds of optimizations are evident in mnemonic techniques like the method of loci (*Yates, 1966*), which involve considerable recoding of to-be-learned information (to maximize distinctiveness of stored memories) and also structured cuing strategies (to ensure that these distinctive memory traces can be found after they are stored). We think that the kinds of policies explored in this paper (e.g., retrieving more when uncertain, encoding more at the end of an event) fall more on the 'automatic' end of the spectrum, as evidenced by the fact that they require no special training and are deployed even in incidental learning situations (e.g., while people are watching a movie, without specifically trying to remember it; *Chen et al., 2016*; *Baldassano et al., 2017*). As such, these policies seem very different from more complex and deliberate kinds of mnemonic strategies like method of loci that require special training. However, we think that it is best to view our 'simple' policies and more complex strategies as falling on a continuum. While the policies we discuss may be deployed automatically in adults, our simulations show that at least some of these policies (e.g., modulating episodic retrieval based on predictive certainty) can be learned through experience, and indeed these strategies might not (yet) be automatic in young children. Furthermore, in principle, there is nothing stopping a model like ours from learning more elaborate strategies given the right kinds of experience and a rich enough action space. Expanding the space of 'memory use policies' for our model and exploring how these can be learned is an important future direction for this work (for a resource-rational approach to memory search, see *Zhang et al., 2021*).

Lastly, although we have focused on cognitive modeling in this paper, we think that some of our results have implications for machine learning more broadly. For example, most memory-augmented neural networks used in machine learning encode at each time point (*Graves et al., 2014*; *Graves et al., 2016*; *Ritter et al., 2018*; *Pritzel et al., 2017*). Our results provide initial evidence that taking episodic 'snapshots' too frequently can actually harm performance. Future work can explore the circumstances under which more selective encoding and retrieval policies might lead to improved performance on machine learning benchmarks. Based on our simulations, we expect that these selective policies will be most useful when there is a substantial risk of recalling lure memories that lead to incorrect predictions, and a substantial cost associated with making these incorrect predictions.

## Summary

The modeling work presented here builds on a wide range of research showing that episodic memory is a resource that the brain can flexibly draw upon to solve tasks (see, e.g., *Shohamy and Turk-Browne, 2013*; *Palombo et al., 2019*; *Palombo et al., 2015*; *Bakkour et al., 2019*; *Biderman et al., 2020*). This view implies that, in addition to studying episodic memory using tasks that probe this system in isolation, it is also valuable to study how episodic memory is used in more complex situations, in concert with working memory and semantic memory, to solve tasks and make predictions. To engage with findings of these sort, we have leveraged advances in AI that make it possible for models to learn how to use episodic memory – our simulations provide a way of predicting how episodic memory should be deployed to obtain rewards, as a function of the properties of the learning environment. While our understanding of these more complex situations is still at an early stage, our hope is that this model (and others like it, such as the model by *Ritter et al., 2018*) can spur a virtuous cycle of predictions, experiments, and model revision that will bring us to a richer understanding of how the brain uses episodic memory.

## Methods

### Episodic retrieval

Episodic retrieval in our model is content-based. The retrieval process returns a weighted average of all episodic memories, where the weight of each memory is equal to its activation; to calculate the activation for each memory, the model executes an evidence accumulation process using a leaky competing accumulator (LCA; *Usher and McClelland, 2001*), which has been used in other memory models (e.g., *Sederberg et al., 2008*; *Polyn et al., 2009*). The evidence for a given episodic memory is the cosine similarity between that memory and the current neocortical pattern (the cell state of the LSTM). Hence, memories that are similar to the current neocortical pattern will have a larger influence on the pattern that gets reinstated.

The evidence accumulation process is governed by the episodic memory gate (EM gate) and the level of competition across memories (*Figure 1C*), which are stored separately from each other (*Figure 1B*). The EM gate is controlled by the neocortical network (*Figure 1A and C*). The EM gate, in turn, controls whether episodic retrieval happens – a higher EM gate value increases the activation of all memories, and setting the EM gate value to zero turns off episodic retrieval completely (see *Appendix 4* for discussion of other ways that gating can be configured). The level of competition (i.e., lateral inhibition) adjusts the contrast of activations across all memories; making the level of competition higher or lower interpolates between one-winner-take-all recall versus recalling an average of multiple memories. In all of our simulations, we set the level of competition to be well above zero (0.8, to be exact), given the overwhelming evidence that episodic retrieval is competitive (*Anderson and Reder, 1999*; *Norman and O'Reilly, 2003*; *Norman, 2010*).

Note that, instead of optimizing the LCA parameters to fit empirical results (e.g., as in the work by *Polyn et al., 2009*), we use a neural network that learns to control the level of the EM gate value. As described below, in the *Model training and testing* section, the model's goal is to maximize reward by making correct predictions and avoiding incorrect predictions; the network learns a policy for setting the EM gate value that maximizes the reward it receives. We made several simplifications to the original LCA – in our model, the LCA (1) has no leak; (2) has no noise; and (3) uses the same EM gate value and competition value for all accumulators.

### Episodic retrieval - detail

At time $t$, assume the model has $n$ memories. The model first computes the evidence for all of the memories. The evidence for the $i$-th memory, $m_i$, is the cosine similarity between the current LSTM cell state pattern $c_t$ and that episodic memory – which is a previously saved LSTM cell state pattern. We denote the evidence for the $i$-th memory as $x_i$:

$$x_i = \text{cosine}\left(c_t, m_i\right)$$

The $x_i$, for all $i$, are the input to the evidence accumulation (LCA) process used in our model; the evidence accumulation process has a timescale $\tau$ that is faster than $t$, such that the accumulation

process runs to completion within a single time point of the neocortical model. The computation at time $\tau$ (for $\tau > 0$) is governed by the following formula:

$$w_\tau^i = \text{relu}\left(\alpha \mathrm{x}_\mathrm{i} - \beta \sum_{\mathrm{j} \neq \mathrm{i}} \mathrm{w}_\tau^\mathrm{j}\right)$$

$w_\tau^i$ is the activation value for the $i$-th memory at time $\tau$; these activations are set to zero initially ($w_0^i = 0$, for all $i$). The activation for the $i$-th memory is positively related to its evidence, $x_i$, and is multiplicatively modulated by $\alpha$, the EM gate value. The $i$-th memory also receives inhibition from all of the other memories, where the level of inhibition is modulated by the level of competition, $\beta$. Finally, the retrieved item at time $t$, denoted by $\mu_t$, is a combination of all memories, weighted by their activation:

$$\mu_t = \sum_{i=1}^{n} w_i m_i$$

## Model training and Testing

### Model training

Before the model is used to simulate any particular experiment, it undergoes a *meta-training* phase that is meant to reflect the experience that a person has prior to the experiment. The goal of this meta-training phase is to let the model learn (1) the structure of the task – how situation features control the transition dynamics across states; and (2) a policy for retrieving episodic memories and for making next-state predictions that maximizes the reward it receives. For every epoch of meta-training, it is trained for all three conditions (recent memory, distant memory, and no memory).

The model is trained with reinforcement learning. Specifically, the model is rewarded/penalized if its prediction about the next state is correct/incorrect. The model also has the option of saying 'don't know' (implemented as a dedicated output unit) when it is uncertain about what will happen next; if the model says 'don't know', the reward is zero. The inclusion of this 'don't know' unit is what motivated us to use reinforcement learning as opposed to purely supervised training. We expected that the optimal way to deploy this action would vary in complex ways as a function of different parameters (e.g., the penalty on incorrect responding), and we did not want to presume that we knew what the best policy would be. Consequently, we opted to let the model learn its own policy for using the 'don't know' action, rather than imposing a policy through supervised training (see *Appendix 5* for results from a variant of the model where we omit the 'don't know' unit and train the model in a purely supervised fashion; these changes make the model less patient – i.e., retrieval takes place earlier in the sequence – but otherwise the results are qualitatively unchanged).

The model is trained with the advantage actor-critic (A2C) objective (*Mnih et al., 2016*). At time $t$, the model outputs its prediction about the next state, $\hat{s}_{t+1}$, and an estimate of the state value, $v_t$. After every event (i.e., a sequence of states of length $T$), it takes the following policy gradient step to adjust the connection weights for all layers, denoted by $\theta$:

$$\nabla_\theta J(\theta) = \nabla\left[\sum_{t=0}^{T} \log \pi_\theta\left(\hat{s}_{t+1}|s_t\right)\left(r_t - v_t\right) - \eta H\big(\pi_\theta(\hat{s}_{t+1}|s_t)\big)\right]$$

where $J$ is the objective function with respect to $\theta$ (the network weights). $\pi_\theta$ is the policy represented by the network weights. $H$ is the entropy function that takes a probability distribution and returns its entropy. $\eta$ controls the strength of entropy regularization.

The objective function $J$ makes rewarded actions (next-state predictions) more likely to occur; the above equation shows how this process is modulated by the level of reward prediction error – measured as the difference between the predicted value, $v_t$, versus the reward at time $t$, denoted by $r_t$. We also used entropy regularization on the network output (*Grandvalet and Bengio, 2006*; *Mnih et al., 2016*) to encourage exploration in the early phase of the training process – with a positive $\eta$, the objective will encourage weight changes that lead to higher entropy distributions over actions.

We used the A2C method (*Mnih et al., 2016*) because it is simple and has been widely used in cognitive modeling (*Ritter et al., 2018*; *Wang et al., 2018*). Notably, there is also evidence that an actor-critic style system is implemented in the neocortex and basal ganglia (*Takahashi et al., 2008*).

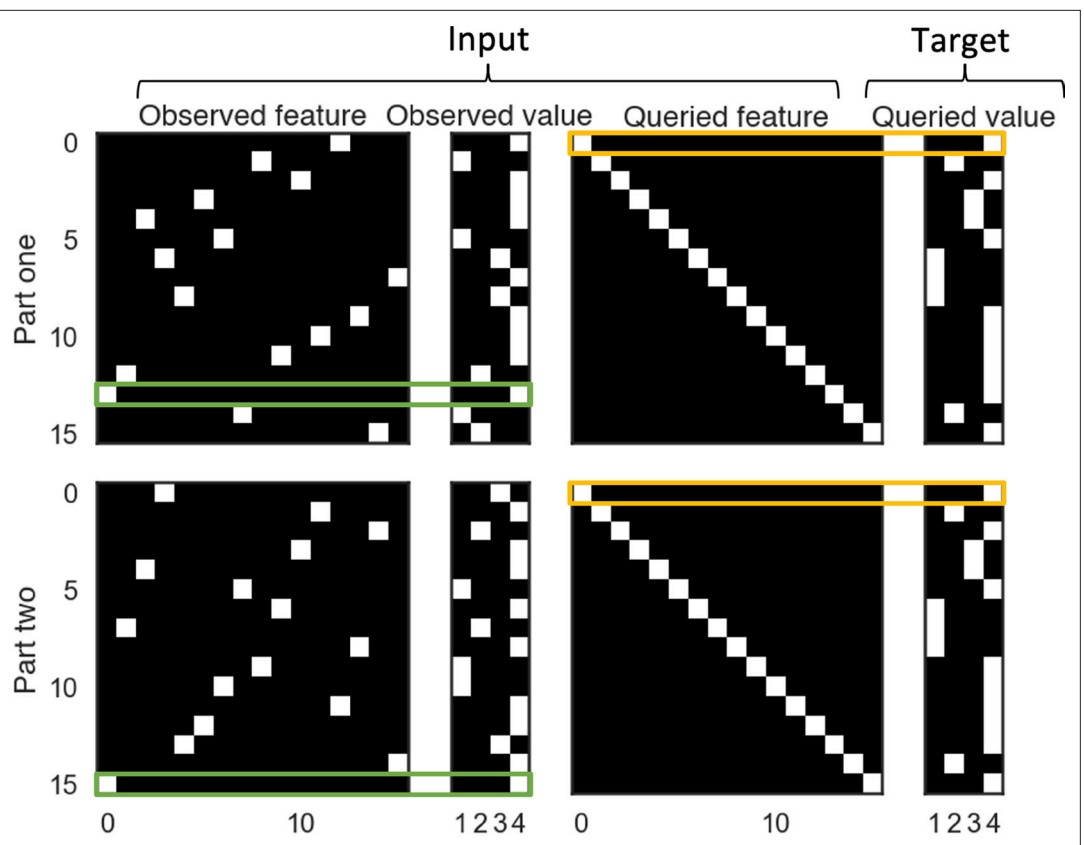

**Figure 5.** The stimulus representation for the event processing task. In the event processing task, situation features are observed in different, random orders during part 1 and part 2, but queries about those features are presented in the same order during part 1 and part 2. The green boxes in panel A indicate time points where the model observed the value of the first feature (time point 13 during part 1, and time point 15 during part 2). The yellow boxes indicate time points where the model was queried about the value of the first feature (time point zero during both part 1 and part 2).

Since pure reinforcement learning is not data-efficient enough, we used supervised initialization during meta-training to help the model develop useful representations (*Misra et al., 2017*; *Nagabandi et al., 2018*). Specifically, the model is first trained for 600 epochs to predict the next state and to minimize the cross-entropy loss between the output and the target. During this supervised pre-training phase, the model is only trained on the recent memory condition and the episodic memory module is turned off, so this supervised pre-training does not influence the network's retrieval policy. Additionally, the 'don't know' output unit is not trained during the supervised pre-training phase – as noted above, we did this because we want the model to learn its own policy for saying 'don't know', rather than having one imposed by us. Next, the model is switched to the advantage actor-critic (A2C) objective (*Mnih et al., 2016*) and trained for another 400 epochs, allowing all weights to be adjusted. The number of training epochs was picked to ensure the learning curves converge. For both the supervised-pretraining phase and the reinforcement learning phase, we used the Adam optimizer (*Kingma and Ba, 2014*) with learning rate of 7e-4 (for more details, please refer to *Appendix 7*).

### Stimulus representation

At time $t$, the model observes a situation feature, and then it gets a query about which state will be visited next. Specifically, the input vector at time $t$ has four components (see *Figure 5*): (1) The observed situation feature (sticking with the example in *Figure 2*, this could be something like 'weather') is encoded as a $T$-dimensional one-hot vector. $T$ is the total number of situation features, which (in most simulations) is the same as the number of time points in the event. The $t$-th one-hot indicates the situation feature governing the transition at time $t$. (2) The value of the observed situation

feature (e.g., learning that the weather is sunny) is encoded as a B-dimensional one-hot vector, where $B$ is the number of possible next states at time $t$. (3) The queried situation feature is encoded as another $T$-dimensional one-hot vector (note that querying the model about the value of the feature that controls the next-state transition is equivalent to querying the model about the next state, given that there is a 1-to-1 mapping between feature values and states within a time point; see *Figure 2A*). (4) Finally, the model also receives the current penalty level for incorrect predictions as a scalar input, which can change across trials. Overall, the input vector at time $t$ is $2T + B + 1$ dimensional. At every time point, there is also a target vector of length $B$ that specifies the value of the queried feature (i.e. the 'correct answer' that the model is trying to predict). The model outputs a vector of length $B + 1$: The first $B$ dimensions correspond to specific predictions about the next state, and the last output dimension corresponds to the 'don't know' response.

In our simulation, the length of an event is 16 time points (i.e., $T$ = 16), and the number of possible states at each time point is 4 (i.e., $B$ = 4). Hence the chance level for next-state prediction is 1/4. *Figure 5* illustrates the stimuli provided to the model for a single example trial. Note that the queries (about the next state) are always presented in the same order, so there is a diagonal on the queried feature matrix. This captures the sequential structure of events (e.g., ordering food always happens before eating the food). However, the order in which the situation features are observed is random. As a result, sometimes a feature is queried after it was observed, in which case the model can rely on its working memory to produce the correct prediction, and sometimes a feature is queried before it was observed, in which case the model needs to use episodic memory (if a relevant memory is available) to produce the correct prediction.

As discussed above, the input vector specifies the level of penalty (for incorrect prediction) for the current trial. During meta-training, the penalty value was randomly sampled on each trial from the range between 0 and 4. During meta-testing, we evaluated the model using a penalty value of 2 (the average of the penalty values used during training). To understand the effect of penalty on retrieval policy, we also compared the timing of recall in the model when the penalty during meta-testing was low (penalty = 0) vs. high (penalty = 4; *Figure 3F*).

In our simulations, during meta-training, the model only got to observe 70% of the features of the ongoing situation during part 1 of the sequence. This was operationalized by giving each feature a 30% probability of being removed during part 1; for time points where the to-be-observed feature was removed, the model observed a zero vector instead. This 'feature removal' during part 1 of the sequence made the task more realistic, since – in general – past information does not fully inform what will happen in the future (during meta-testing, we did not remove any observations during part 1; this makes the results graphs easier to interpret, but has no effect on the conclusions reported here).

Finally, we wanted to make sure the model could adjust its retrieval time flexibly, instead of learning to always retrieve at a fixed time point (e.g., always retrieve at the third time point). Therefore, during training, we delayed the prediction demand by a random number of time points (from 0 to 3). For example, if the amount of delay was two in a given trial, then the model observed two situation features before it received the first query.

## Model Testing

During meta-testing (i.e., model evaluation; when simulating a particular experiment), the weights of the neocortical part of the model (i.e., all weights pertaining to the LSTM, decision layer, and EM gate) were frozen, but the model was allowed to form new episodic memories. In any given trial (where the model observed several events), new learning of information completely relied on working memory (i.e., model's recurrent dynamics), episodic memory in the episodic module, and semantic memory encoded in the (frozen) neocortical connection weights (instantiating the model's knowledge of transitions between states and how these transitions are controlled by situation features). The results shown in all the simulations were obtained by testing the model with new, randomly-generated events, after the initial meta-training phase. While it is theoretically possible that these test events could duplicate events that were encountered during meta-training, exact repeats will be very rare due to the combinatorics of the stimuli (as noted earlier, there are $4^{16}$ possible sequences of states within an event). For more information on model parameters, see *Appendix 7*.

## Decoding the working memory state

In *Figure 2C*, we used a decoding approach to track what information the model was maintaining in working memory over time while it processes an event. This approach allowed us to assess the model's ability to hold on to observed features after they were observed, and also to detect when features were retrieved from episodic memory and loaded back up into working memory. Our use of decoders here is analogous to the widespread use of multivariate pattern analysis (MVPA) methods to decode the internal states of participants from neuroimaging data (*Haxby et al., 2001*; *Norman et al., 2006*; *Lewis-Peacock and Norman, 2014*) – the only difference is that, here, we applied the decoder to the internal states of the model instead of applying it to brain data.

Specifically, we trained classifiers on LSTM cell states during part 1 to decode the feature values over time. Each situation feature was given its own classifier (logistic regression with L2 penalty). For example, if 'weather' was one of the situation features, we would train a dedicated 'weather' classifier that takes the LSTM cell state and predicts the value of the weather feature for a given time point. To set up the targets for these classifiers for part 1, we labeled all time points before the model observed the feature value as 'don't know'. After a feature value was revealed, we labeled that time point and the following time points with the value of that feature (e.g., if the weather feature value was observed to be 'rainy' on time point 4, then time point four and all of those that followed until the end of part 1 of the sequence were labeled with the value 'rainy'). For part 2 data, we assumed all features were reinstated to the model's working memory state after the EM gate value peaked. This labeling scheme assumes that (1) observed features are maintained in working memory and (2) episodic recall brings back previously encoded information. These assumptions can be tested by applying the classifier to held-out data. When decoding working memory states during part 1 of the sequence, we used a fivefold cross-validation procedure, and picked the regularization parameter with an inner-loop cross-validation. All results were generated using held-out test sets. The average decoding accuracy was 91.58%. Note that, as mentioned above, there is no guarantee that features observed earlier in the sequence will be maintained in the model's working memory. As such, below-ceiling decoding accuracy could reflect either (1) failure to accurately decode the contents of working memory or (2) the decoder accurately detecting a working memory failure (i.e., that the feature in question has 'dropped out' of the model's working memory, despite having been observed earlier in the sequence).

## Code

Github repo: https://github.com/qihongl/learn-hippo (*Lu, 2022*; copy archived at swh:1:rev:6a4a1be4fd6780d4c8413ffc6b1facade4741135).

## Acknowledgements

This work was supported by a Multi-University Research Initiative grant awarded to KAN and UH (ONR/DoD N00014-17-1-2961). We are grateful for the feedback we have received from members of the Princeton Computational Memory Lab, the Hasson Lab, and the labs of our MURI collaborators Charan Ranganath, Lucia Melloni, Jeffrey Zacks, and Samuel Gershman.

## Additional information

### Funding

| Funder | Grant reference number | Author |
|---|---|---|
| Office of Naval Research | Multi-University Research Initiative Grant ONR/DoD N00014-17-1-2961 | Kenneth A Norman Uri Hasson |

The funders had no role in study design, data collection and interpretation, or the decision to submit the work for publication.

## Author contributions
Qihong Lu, Conceptualization, Formal analysis, Investigation, Methodology, Software, Visualization, Writing – original draft, Writing – review and editing; Uri Hasson, Conceptualization, Funding acquisition, Supervision, Writing – review and editing; Kenneth A Norman, Conceptualization, Funding acquisition, Methodology, Project administration, Supervision, Writing – review and editing

## Author ORCIDs
Qihong Lu ⓘ http://orcid.org/0000-0002-0730-5240
Kenneth A Norman ⓘ http://orcid.org/0000-0002-5887-9682

## Decision letter and Author response
Decision letter https://doi.org/10.7554/eLife.74445.sa1
Author response https://doi.org/10.7554/eLife.74445.sa2

---

# Additional files

## Supplementary files
• Transparent reporting form

## Data availability
The code is made publicly available here in a git repo: https://github.com/qihongl/learn-hippo (copy archived at swh:1:rev:6a4a1be4fd6780d4c8413ffc6b1facade4741135). Users can also use this Code Ocean capsule to play with one example model to qualitatively replicate some results: https://code-ocean.com/capsule/3639589/tree.

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

# Appendix 1

## The internal representation of the decision layer

To explore how neural activity patterns in the decision layer differed as a function of certainty, we plotted the activity patterns as a function of the action taken by the model (i.e., whether it predicted one of the four upcoming states, or whether it used the 'don't know' response). *Appendix 1—figure 1* shows the results of this analysis: Uncertain states are approximately clustered near the center of the activation space (with a lower L2 norm) while other responses are farther away, which indicates that uncertainty in our model is represented by the absence of evidence towards any particular choice. Importantly, this difference in activity patterns is not built-in to the model – it simply emerges during training.

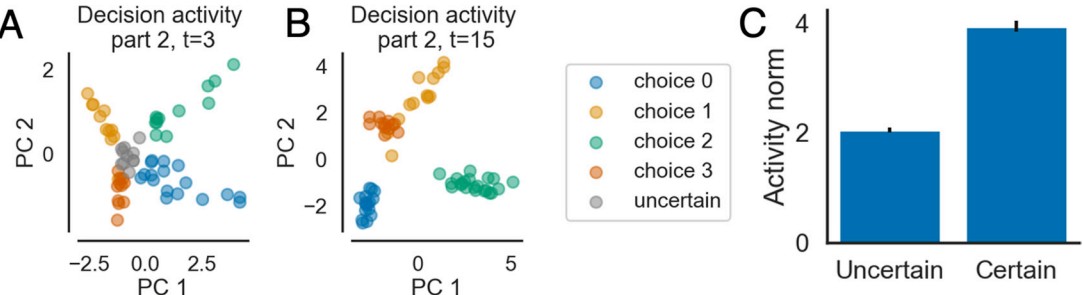

**Appendix 1—figure 1.** How certainty is represented in the model's activity patterns. Panels A and B show the neural activity patterns from the decision layer in the distant memory (DM) condition, projected onto the first two principal components. Each point corresponds to the pattern of neural activity for a trial at a particular time point. We colored the points based on the output (i.e., 'choice') of the model, which represents the model's belief about which state will happen next. Patterns that subsequently led to 'don't know' responses are colored in grey. Panel A shows an early time point with substantial uncertainty (a large number of 'don't know' responses). Panel B shows the last time point of this event, where the model has lower uncertainty. Panel C shows the average L2 norm of states that led to 'don't know' responses (uncertain) versus states that led to specific next-state predictions (certain); the errorbars indicate 1SE across 15 models. States corresponding to 'don't know' responses are clustered in the center of the activation space, with a lower L2 norm.

## Appendix 2

### Effects of event similarity on retrieval policy

In this simulation, we studied how the similarity of event memories in the training environment affects retrieval policy. To manipulate memory similarity, we varied the proportion of shared situation feature values across events during training. In the low-similarity condition, the similarity between the distractor situation (i.e., situation A; see *Figure 2* in the main text) and the target situation was constrained to be less than 40%, so target memories and lures were relatively easy to distinguish. In the high-similarity condition, the similarity between the distractor situation and the target situation was constrained to fall between 35% and 90%. We used a rejection sampling approach to implement these similarity bounds – during stimulus generation, we kept generating distractor situations until they fell within the similarity bounds with respect to the target sequence. Otherwise, the simulation parameters were the same as the parameters that were used in the main text.

In the high-similarity condition, target and lure memories were more confusable, and thus the risk of lure recall was higher. In light of this, we expected that the model would adopt a more conservative retrieval policy (i.e., retrieving less) in the high-similarity condition. We also expected that this effect would be stronger when the penalty is high; when the penalty is low, there is less of a cost for recalling the lure memory, and thus less of a reason to refrain from episodic retrieval in the high-similarity condition.

We compared the model's behavior as a function of penalty and similarity. For the penalty manipulation, each model was trained on a range of penalty values from 0 to 4, then tested on low (0), moderate (2), and high (4) penalty values. *Appendix 2—figure 1* shows the average level of memory activation in each of the conditions. As expected, memory activation is lower in the high-similarity condition, especially when the penalty is high. Notably, increasing penalty reduces memory activation in the high-similarity condition (where the risk of false recall is high) but it does not have this effect in the low-similarity condition (where the risk of false recall is low).

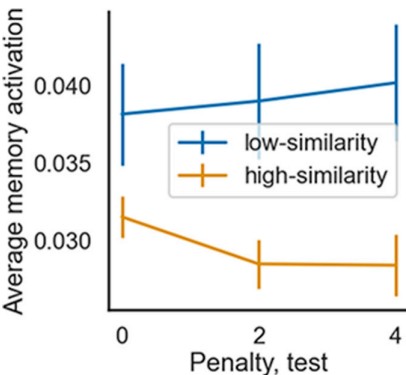

**Appendix 2—figure 1.** Memory activation during part 2 (averaged over time) in the DM condition, for models trained in low vs. high event-similarity environments and tested with penalty values that were low (penalty = 0), moderate (penalty = 2), or high (penalty = 4). The model recalls less when similarity is high (vs. low), and this effect is larger for higher penalty values. The errorbars indicate 1SE across 15 models.

## Appendix 3

### Effects of familiarity on retrieval policy

Prior work has demonstrated that neocortex is capable of computing a scalar *familiarity signal* that discriminates between previously-encountered and novel stimuli (*Yonelinas, 2002*; *Norman and O'Reilly, 2003*; *Holdstock et al., 2002*); likewise, signals can be extracted from models of the hippocampus that discriminate between novel and familiar stimuli (e.g., *Hasselmo and Wyble, 1997*; see *Norman, 2010* for a comparison of the properties of these neocortical and hippocampal signals). In this section, we study how familiarity signals can support episodic retrieval policy. Relevant to this point, several recent studies in humans have found that encountering a familiar stimulus can temporarily shift the hippocampus into a 'retrieval mode' where it is more likely to retrieve episodic memories in response to available retrieval cues (*Duncan et al., 2012*; *Duncan and Shohamy, 2016*; *Duncan et al., 2019*; *Patil and Duncan, 2018*). Here, we assess whether our model can provide a resource-rational account of these 'retrieval mode' findings.

Intuitively, familiarity can guide episodic retrieval policy by providing an indication of whether a relevant episodic memory is available. If an item is unfamiliar, this signals that it is unlikely that relevant episodic memories exist, hence the expected benefit of retrieving from episodic memory is low (if there are no relevant episodic memories, episodic retrieval can only yield irrelevant memories, which lead to incorrect predictions); and if an item is familiar, this signals that relevant episodic memories are likely to exist and hence the benefits of retrieving from episodic memory are higher. These points suggest that the model would benefit from a policy whereby it adopts a more liberal criterion for retrieving from episodic memory when stimuli are familiar as opposed to novel.

To test this, we ran simulations where we presented a 'ground truth' familiarity signal to the model during part 2 of the sequence. The familiarity signal was presented using an additional, dedicated input unit (akin to how we present penalty information to the model). Specifically, during part 2, if the ongoing situation had been observed before (as was the case in the RM and DM conditions), the familiarity signal was set to one. In contrast, if the ongoing situation was novel (as was the case in the NM condition), then the familiarity signal was set to negative one. Before part 2, the familiarity signal was set to zero (an uninformative value). Other than these changes, the parameters of this simulation were the same as the other simulations. The model was tested on penalty value of 2 – the average of the training range. Note that our treatment of the familiarity signal here deliberately glosses over the question of how this signal is generated, as this question is addressed in detail in other models (e.g., *Norman and O'Reilly, 2003*; *Hasselmo and Wyble, 1997*); our intent here is to understand the consequences of having a familiarity signal (however it might be generated) for the model's episodic retrieval policy.

*Appendix 3—figure 1* and *Appendix 3—figure 2* illustrate prediction performance, memory activation, and EM gate values for models with and without the familiarity signal. When the model has access to a veridical familiarity signal ( + 1 for RM and DM, –1 for NM), it opens the EM gate immediately and strongly in the DM condition (*Appendix 3—figure 2D*), leading to higher activation of both the target memory and the lure (*Appendix 3—figure 2A*) in the DM condition, relative to models without the familiarity signal (*Appendix 3—figure 2B*). Behaviorally, models with the familiarity signal show both a higher correct prediction rate and a slightly higher error rate in the DM condition, compared to models without the familiarity signal (*Appendix 3—figure 1A* vs. B). This slight increase in errors occurs because, when the model retrieves immediately from episodic memory during part 2, the model (in some cases) has not yet made enough observations to distinguish the target and the lure. In the NM condition, with the familiarity signal, the model keeps the EM gate almost completely shut (*Appendix 3—figure 2D*). Consequently, the level of memory activation stays very low in the NM condition (*Appendix 3—figure 2A*), which reduces the error rate in the NM condition to zero (*Appendix 3—figure 1A*). The RM condition is an interesting case: Previously (see *Figure 3* in the main text), we found that the model refrained from episodic memory retrieval in the RM condition; we found that the same pattern is present here, even when we make a familiarity signal available to the model: EM gate and memory activation values are both very low (*Appendix 3—figure 2A and D*), similar to models without access to the familiarity signal (*Appendix 3—figure 2B and E*). This shows that model does not always retrieve from episodic memory when given a high familiarity signal – in this case, the presence of relevant information in working memory (which suppresses episodic retrieval) 'overrides' the presence of the familiarity signal (which enhances episodic retrieval in the DM condition).

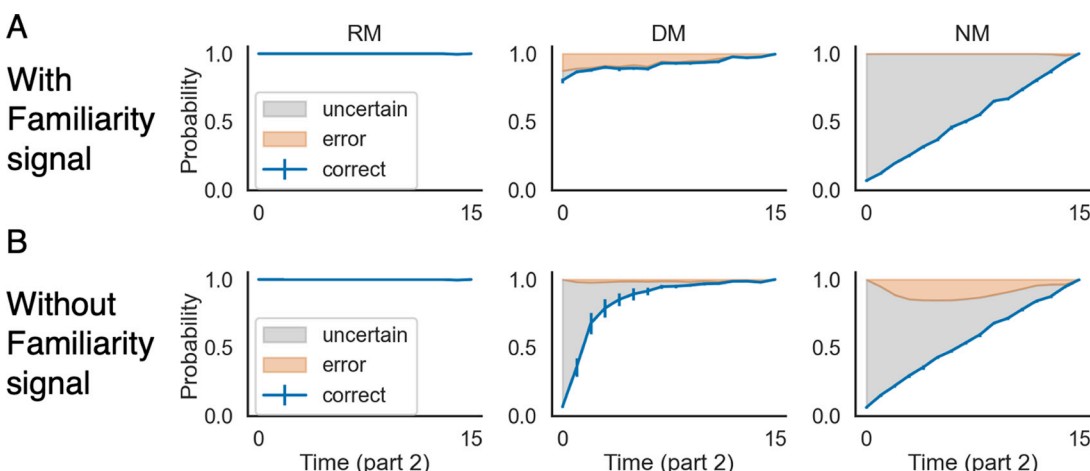

**Appendix 3—figure 1.** The familiarity signal can improve prediction. Next-state prediction performance for models with (**A**) vs. without (**B**) access to the familiarity signal. With the familiarity signal (**A**), the model shows (1) higher levels of correct prediction in the DM condition, and (2) a reduced error rate in the NM condition. The errorbars indicate 1SE across 15 models.

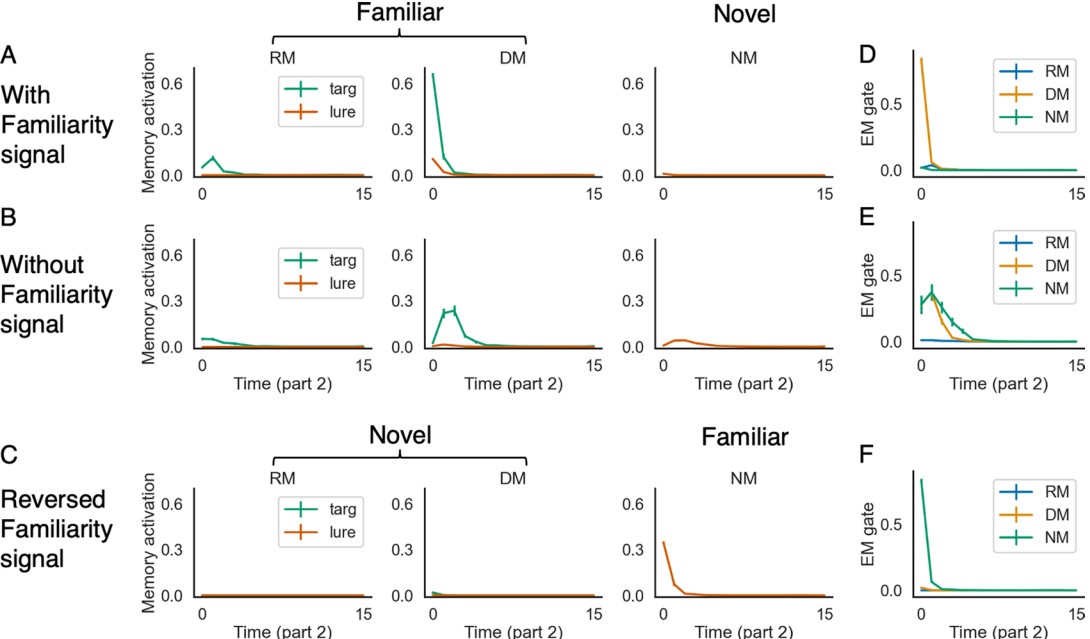

**Appendix 3—figure 2.** Episodic retrieval is modulated by familiarity. This figure shows the memory activation and EM gate values over time for three conditions: (1) with the familiarity signal (**A, D**), (2) without the familiarity signal (**B, E**), and (3) with a reversed (opposite) familiarity signal at test (**C, F**). With the familiarity signal (**A**), the model shows higher levels of recall in the DM condition, and suppresses recall even further in the NM condition, compared to the model without the familiarity signal (**B**). This is due to the influence of the EM gate – the model with the familiarity signal retrieves immediately in the DM condition, and turns off episodic retrieval almost completely in the NM condition (**D**). Note also that levels of episodic retrieval in the RM condition stay low, even with the familiarity signal (see text for discussion). Finally, parts **C** and **F** show that reversing the familiarity signal at test suppresses recall in the DM condition and boosts recall in the NM condition. The errorbars indicate 1SE across 15 models.

Finally, we can trick the model into reversing its retrieval policy by reversing the familiarity signal at test (*Appendix 3—figure 2C and F*). In this condition, the (reversed) signal indicates that the ongoing situation is novel (–1) in the RM and the DM condition, and the ongoing situation is familiar (+1) in the NM condition. As a result, the model suppresses episodic retrieval in the RM and DM conditions, and recalls lures in the NM condition.

Overall, the results of this simulation show that our model is able to use a familiarity signal to inform its retrieval policy in the service of predicting upcoming states. Consistent with empirical results (*Duncan et al., 2012*; *Duncan and Shohamy, 2016*; *Duncan et al., 2019*; *Patil and Duncan, 2018*), we found that the model retrieves more from episodic memory when the ongoing situation is familiar, unless the model has low uncertainty about the upcoming state. These modeling results provide a resource-rational account of why familiarity leads to enhanced episodic retrieval.

## Appendix 4

### Alternative configurations of episodic memory gating

In the simulations described in the main text, the EM gate controls the input into the EM system. An alternative way of accomplishing gating is to place the gate *after* the EM module (LCA), so it controls the flow of activation from the EM module back into the LSTM. *Appendix 4—figure 1* illustrates the differences between these configurations; for convenience, we will use 'post-gating' to refer to the latter mechanism and 'pre-gating' to refer to the mechanism used in the simulations described in the main text. As noted in the *Discussion*, the primary consequence of having the gate on the output side is that the gate can be controlled based on information coming out of the hippocampus, in addition to all of the neocortical regions that are used to control the gate in our pre-gating model. The post-gating mechanism has been more widely used in machine learning (*Ritter et al., 2018*; *Ritter, 2019*; *Pritzel et al., 2017*) because it is more powerful – since the gating function has access to activated episodic memories in the LCA, the model can close/open the gate depending on the properties of these activated memories.

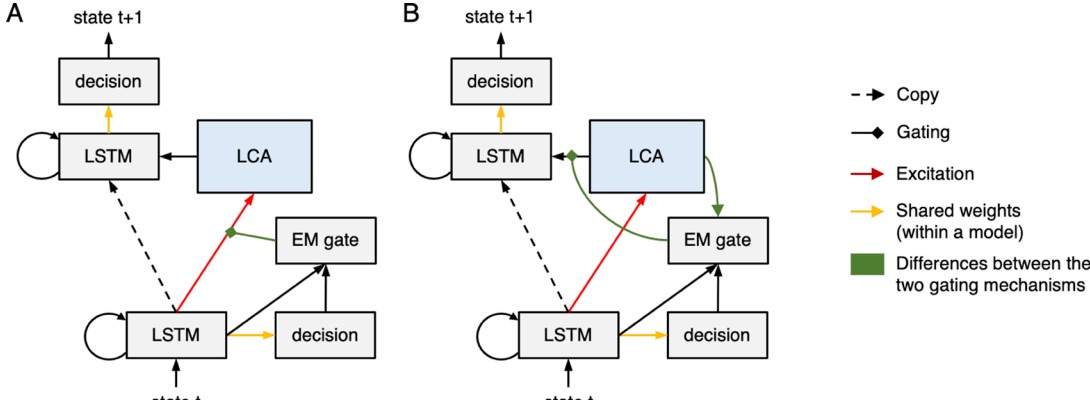

**Appendix 4—figure 1.** Unrolled network diagrams for the pre-gating (**A**) versus the post-gating (**B**) models. The EM gate in the pre-gating model controls the degree to which stored memories are activated within the LCA module, but does not control the degree to which the activated memories are transmitted to the neocortex. By contrast, the EM gate in the post-gating model controls the degree to which activated memories in the LCA module are transmitted to the neocortex, but it does not control how these memory activations are computed in the first place.

Since it is still unclear what kinds of episodic memory gating are implemented in the brain (see below for further discussion), we experimented with both mechanisms. We focused on the pre-gating model in the main text since it involves fewer assumptions – critically, it does not assume that the gating mechanism has access to the content of memories that are activated within the hippocampus. That said, the key results for the pre-gating model, reported in the main text, qualitatively hold for the post-gating model (*Appendix 4—figure 2*). In particular, the post-gating model also (1) retrieves much more from episodic memory in the DM condition, compared to the other two conditions (*Appendix 4—figure 2A, B and C*); (2) retrieves more when it is uncertain about the upcoming state (*Appendix 4—figure 2D*); (3) delays its recall time when the penalty is higher (*Appendix 4—figure 2E*); (4) adjusts its EM gate value as a function of the schema strength in a way that is similar to the pre-gating model (*Appendix 4—figure 2F*); and( 5) shows the effect that midway-encoded memories hurt next-state prediction performance (*Appendix 4—figure 2G and H* – note that this also holds true when midway-encoded memories are present during meta-training). Importantly, while the aforementioned patterns replicate across the models, the results are not exactly the same – the retrieval policy for the post-gating model is often more flexible (i.e., it can adapt better to current conditions), since its EM gate can be controlled by the output of the EM module (in addition to the output of other neocortical regions). For example, in the post-gating model, the EM gate layer of the neocortical network is able to detect that relevant memories are not present in the NM condition, and it adapts to this by setting the EM gate to a lower value in the NM condition than the DM condition (*Appendix 4—figure 2C*) – that is, it learns to suppress retrieval when no memories are coming to mind. By contrast, the pre-gating model actually shows the opposite pattern – here, the EM gate layer can not detect the absence of relevant memories in the NM condition, but it *can* detect higher overall levels of uncertainty in the NM condition than the DM condition, which leads

it to set the EM gate to a slightly higher value in the NM condition than the DM condition (see *Figure 3C* in the main text).

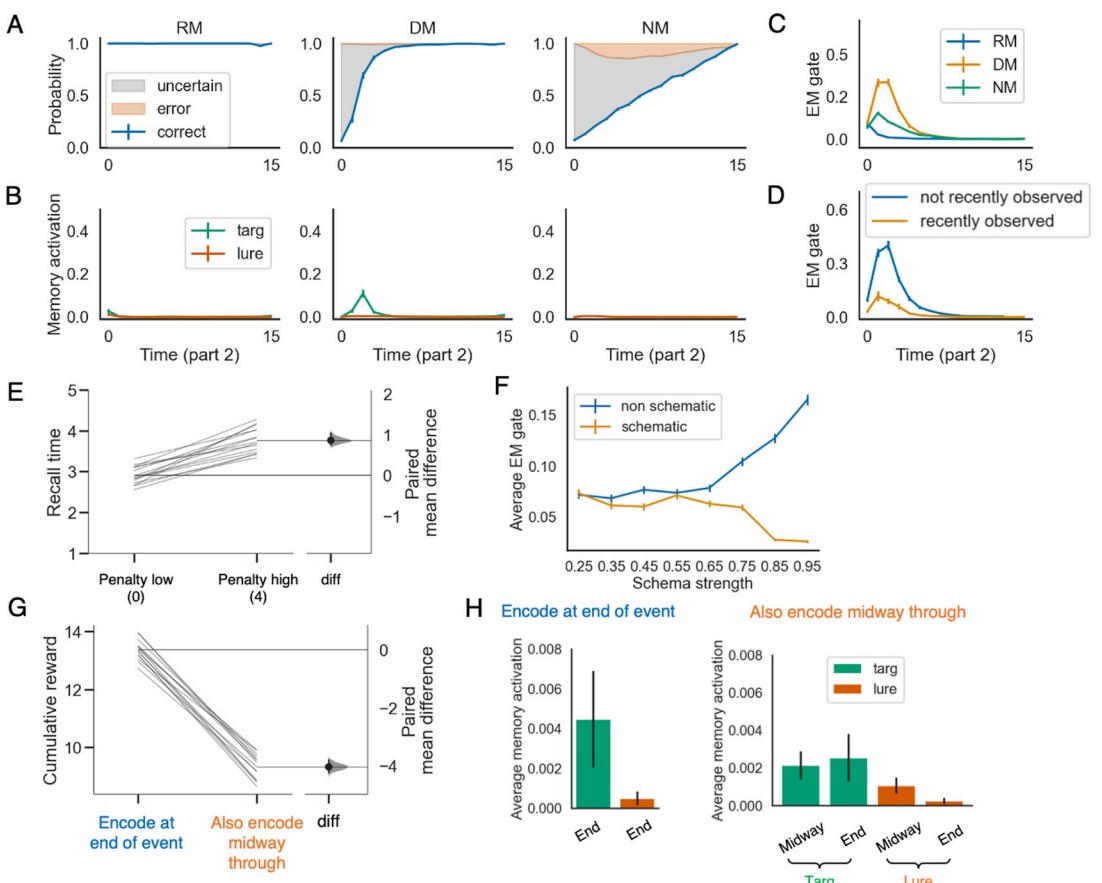

**Appendix 4—figure 2.** The post-gating model qualitatively replicates key results obtained from the pre-gating model (compare to *Figures 3 and 4* in the main text). See text in this appendix for discussion. The errorbars indicate 1SE across 15 models.

One exciting future direction is to experimentally investigate how episodic memory gating works in the brain. The pre-gating and post-gating models make different predictions about the hippocampal activity: The post-gating model predicts that candidate episodic memory traces should be activated in the hippocampus at each time point; sometimes these activated traces are blocked (by the gate) from being transmitted to neocortex, and sometimes they are allowed through. The pre-gating model predicts that activation of episodic memory traces in the hippocampus will distributed more sparsely in time; on time points when the gate is closed, no activation should be transmitted from neocortex to hippocampus, resulting in reduced activation of hippocampal memory traces (although there might be activation of these traces via recurrence within the hippocampus). Putting these points together, the pre-gating model appears to predict a large difference in hippocampal activation patterns as a function of whether the gate is closed or open; by contrast, the post-gating model appears to predict a smaller difference in hippocampal activation patterns as a function of whether the gate is closed or open.

However, this logic is complicated by the fact that the hippocampus is connected in a recurrent 'big loop' with neocortex (*Schapiro et al., 2017*; *Kumaran and Maguire, 2007*; *van Strien et al., 2009*; *Koster et al., 2018*) – in the post-gating model, even if the inputs to the hippocampus are the same when the gate is open vs. closed, the outputs to neocortex will be different, which in turn will affect the inputs (from neocortex) that hippocampus receives on the next time point. Thus, we would eventually expect differences in hippocampal activation in these conditions, even in the post-gate model. This suggests that, while it may be challenging to empirically tease apart the pre-gating and post-gating models, time-resolved methods like ECoG that can (in principle) distinguish between the 'initial wave' of activity hitting the hippocampus after a stimulus and subsequent (recurrent) waves of activity would be most useful for this purpose. We should also note that the

pre-gating and post-gating mechanisms are not mutually exclusive and it is possible that the brain deploys both of them.

## Appendix 5

### Training the model without reinforcement learning

In the simulations shown in the main text, we trained the model using reinforcement learning (after supervised pre-training) and gave the model the option of responding 'don't know', in which case it received no penalty or reward (see *Model training and testing* section above for details). Here, in *Appendix 5—figure 1*, we report the results from a model variant in which the model was trained in an entirely supervised fashion, without the option of responding 'don't know' – on each time point, the model was forced to predict the next state, and weights were adjusted based on the discrepancy between the predicted and actual states.

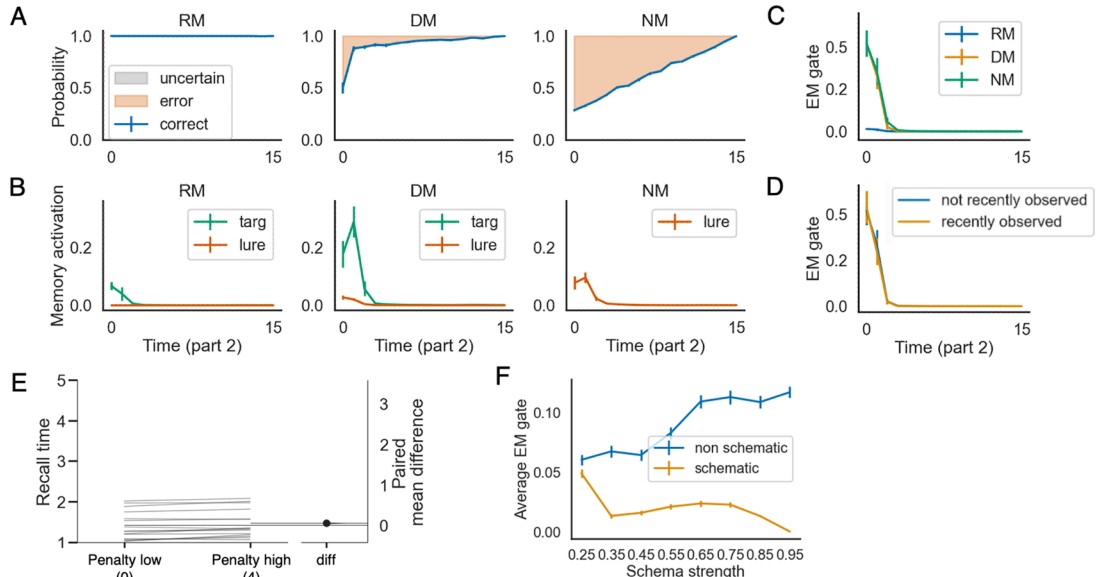

**Appendix 5—figure 1.** Results from a 'no-RL' model that was trained in an entirely supervised fashion, without reinforcement learning and without the option of giving a 'don't know' response – compare to **Figure 3** in the main text; see text in this appendix for discussion. The errorbars indicate 1SE across 15 models.

There are two important observations to make based on the results in *Appendix 5—figure 1*. The first observation is that the model is much less patient (i.e., it retrieves much earlier in part 2) when we take away the option of giving a 'don't know' response. This impatience can be seen by comparing the early time points of *Appendix 5—figure 1C* to the early time points of *Figure 3C* in the main text – EM gate values are much higher at early time points in the no-RL model. It can also be seen by comparing *Appendix 5—figure 1E* to *Figure 3E* in the main text – the average time-to-recall is much lower in the no-RL model. These findings confirm our claim (made in the main text) that the 'don't know' response makes the strategy of waiting to retrieve more viable, by allowing the model to escape being penalized on trials when it is waiting to retrieve from episodic memory.

The second observation is that, even without the option of responding 'don't know', the learned retrieval policy of the no-RL model is still sensitive to certainty. This is shown in *Appendix 5—figure 1B and C*: Just like the model in the main text, the no-RL model recalls less information in the RM condition (when it is more certain about what will happen next) vs. the DM condition. The lack of a difference in EM gate value between 'recently observed' and 'not recently observed' features in *Appendix 5—figure 1D* suggests that the no-RL model might *not* be sensitive to certainty, but this is an artifact of the no-RL model's impatience – the EM gate value is very high for early time points in both conditions, making it harder to observe a difference between conditions; in other simulations (not shown here) where we used a stronger penalty manipulation to disincentivize early retrieval, the difference in recall levels between 'recently observed' and 'not recently observed' features was clearly visible in the no-RL model, reaffirming its sensitivity to certainty.

Taken together, the results from the no-RL model are very useful in clarifying what, exactly, is gained from the use of RL training with a 'don't know' option. In particular: having a 'don't know' response does not *cause* the model to have qualitatively distinct neural states as a function of certainty – these differences (described in Appendix 1 above) exist regardless of 'don't know' training, and can be used by the no-RL model to modulate its retrieval policy. Rather, the effect of RL training with the

'don't know' response is to make the model more patient, by giving it the option of waiting without penalty when it is uncertain.

## Appendix 6

### Simulating inter-subject correlation results from *Chen et al., 2016*

As discussed in the main text, *Chen et al., 2016* found strong hippocampal-neocortical activity coupling measured using inter-subject functional connectivity (ISFC; *Simony et al., 2016*) for DM participants, while the level of coupling was much weaker for participants in the RM and NM conditions (*Chen et al., 2016*). Here, we address some additional findings from this study that used temporal inter-subject correlation (ISC) as a dependent measure; temporal ISC tracks the degree to which the fMRI time series in a particular brain region is correlated across participants (*Hasson et al., 2004*; *Chen et al., 2016*; *Nastase et al., 2019*). Specifically, *Chen et al., 2016* found that – at the start of part 2 – temporal ISC in DMN regions was lower between participants in the DM and RM conditions than between RM participants, suggesting differences in how DM and RM participants were interpreting the story; however, this gap in ISC decreased over the course of part 2, suggesting that these differences in interpretation between DM and RM participants decrease over time (*Appendix 6—figure 1B*). Furthermore, across participants, the degree to which the gap in ISC narrowed during the second half of part 2 was correlated with the amount of hippocampal-neocortical activity coupling at the start of part 2 (*Appendix 6—figure 1C*; *Chen et al., 2016*). Taken together, these findings can be interpreted as showing that hippocampus is consulted more (as evidenced by increased hippocampal-neocortical coupling) in the DM condition (where there are gaps in the situation model at the start of part 2) than the RM condition (where the situation model is more complete); the effect of this increased consultation of the hippocampus is to 'fill in the gaps' and align the interpretations of the DM and RM participants (as evidenced by DM-RM ISC rising to the level of RM-RM ISC).

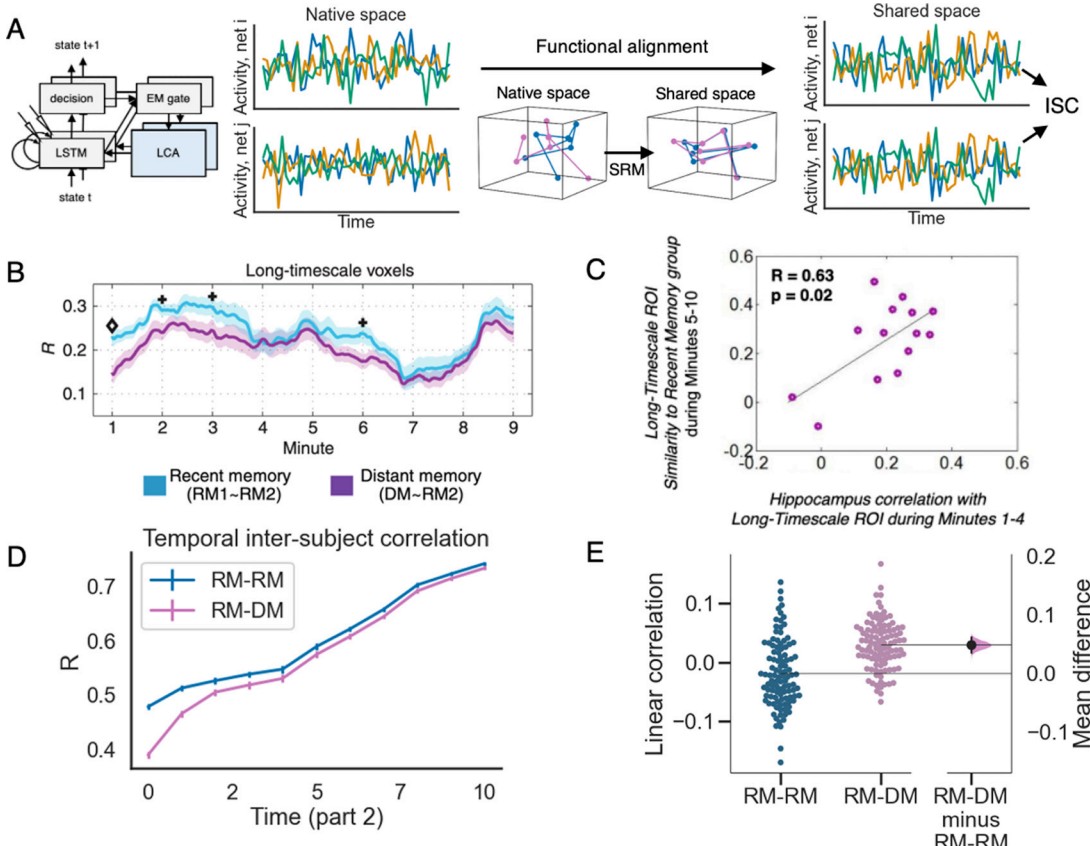

**Appendix 6—figure 1.** Simulating inter-subject correlation results from *Chen et al., 2016*. (**A**) Illustration of how we computed inter-subject correlation (ISC) in the model (see text for details). (**B and C**) show the empirical results from *Chen et al., 2016* (reprinted with permission) and (**D and E**) show model results. (**B**) The sliding-window temporal inter-subject correlation (ISC) over time, during part 2 of the movie. The recent memory ISC, or RM-RM ISC, was computed as the average ISC value between two non-overlapping subgroups of the RM participants. The distant memory ISC, or RM-DM ISC, was computed as the average ISC between one sub-group of RM participants
*Appendix 6—figure 1 continued on next page*

*Appendix 6—figure 1 continued*

and the DM participants. Initially, the RM-DM ISC was lower than RM-RM ISC, but as the movie unfolded, RM-DM ISC rose to the level of RM-RM ISC. (**C**) For the DM participants, the level of hippocampal-neocortical inter-subject functional connectivity at the beginning of part 2 of the movie (minutes 1–4) was correlated with the level of RM-DM ISC later on (minutes 5–10). (**D**) Sliding window temporal ISC in part 2 between the RM models (RM-RM) compared to ISC between the RM and DM models (RM-DM). The convergence between RM-DM ISC and RM-RM ISC shows that activity dynamics in the DM and the RM models become more similar over time (compare to part B of this figure). The errorbars indicate 1SE across 15 models. (**E**) The correlation in the model between memory activation at time $t$ and the change in ISC from time $t$ to time $t + 1$, for the first 10 time points in part 2. Each point is a subject-subject pair across the two conditions. The 95% bootstrap distribution on the side shows that the correlation between memory activation and the change in RM-DM ISC is significantly larger than the correlation between memory activation and the change in RM-RM ISC (see text for details).

To simulate these results, we trained 30 neural networks, then we assigned half of them to the RM condition and half to the DM condition. Next, we performed the temporal ISC analysis used in *Chen et al., 2016* by treating hidden-unit activity patterns as multi-voxel brain patterns. An important technical note is that running ISC across networks requires some form of alignment (i.e., so the time series for corresponding parts of the networks can be correlated). Human fMRI data are approximately aligned across subjects, since brain anatomy is highly similar across people. However, when many instances of the same neural network architecture are trained on the same data, they tend to acquire different neural representations, even though they represent highly similar mathematical functions (*Li et al., 2015*; *Dauphin et al., 2014*; *Meng et al., 2018*). That is, the same input can evoke uncorrelated neural responses across different networks, although they produce similar outputs. For our purpose, this means that directly correlating hidden-layer activity patterns across neural networks will underestimate the similarity of representations across networks. Therefore, to simulate effects involving (human) inter-subject analyses, we need a way to align neural networks.

To accomplish this goal, we used the shared response model (SRM) (*Lu et al., 2018*) – a functional alignment procedure commonly used for multi-subject neuroimaging data (*Chen et al., 2015b*; *Haxby et al., 2011*; *Haxby et al., 2020*). Intuitively, this method applies rigid body transformation to align different network activities into a common space. We have previously shown that neural networks with highly overlapping training experience can be aligned well with SRM (*Lu et al., 2018*). Here, we used the Brain Imaging Analysis Kit (BrainIAK) implementation of SRM (*Kumar et al., 2020b*; *Kumar et al., 2020a*) to align our trained networks before computing ISC (*Appendix 6— figure 1A*).

Our simulation results qualitatively capture the findings from *Chen et al., 2016*. During part 2, DM-RM ISC starts lower than RM-RM ISC, but as the event unfolds, they gradually converge (*Appendix 6—figure 1D*). Moreover, in the DM condition, the level of memory activation at time $t$ is correlated with the increment in DM-RM ISC from time $t$ to time $t + 1$ (*Appendix 6—figure 1E*). As a comparison point, in the RM condition (where the model is not relying on episodic retrieval to fill in gaps in the situation model), memory activation does not correlate with the change in (RM-RM) ISC. Collectively, these results establish that episodic retrieval accelerates the convergence between model activations in the DM and RM conditions.

More generally, this result shows that one can capture inter-subject results with computational models. Experiments using inter-subject analyses and natural stimuli are becoming increasingly popular (*Nastase et al., 2019*; *Sonkusare et al., 2019*; *Hamilton and Huth, 2018*; *Nastase et al., 2020*); our simulation results provide a proof-of-concept demonstration of how computational models of memory can engage with this literature.

## Appendix 7

### Model parameters

We implemented the model in PyTorch (*Paszke et al., 2017*; *Paszke et al., 2019*). The numbers of hidden units for the LSTM layer and the decision layer were 194 and 128, respectively. The level of competition in the LCA module was 0.8. The initial cell state of the LSTM was a random vector ~ isotropic Gaussian(0,.1).

During the meta-training phase, we used the Adam optimizer (*Kingma and Ba, 2014*). The initial learning rate was 7e-4. The learning rate decayed by 1/2 if the average prediction accuracy minus mistakes stayed within 0.1% from the previous best loss for 30 consecutive epochs. The minimal learning rate was 1e-8. We used orthogonal weight initialization with gain of 1 (*Saxe et al., 2014*), and we used supervised initialization for 600 epochs to help the model develop useful representations (*Misra et al., 2017*; *Nagabandi et al., 2018*). During the supervised initialization phase, the model was trained to predict the upcoming state; episodic memory and the 'don't know' unit were turned-off during this phase. After the supervised initialization phase, the model was trained with A2C (*Mnih et al., 2016*) for another 400 epochs. We used entropy regularization with weight of 0.1 to encourage exploration. For every epoch, the model was trained on 256 events.

