## [Editor Report]

This paper addresses an important problem in control of episodic memory. This paper develops a computationally-based proposal about how semantic, working memory, and episodic memory systems might learn to interact so that stored episodic memories can optimally contribute to reconstruction of semantic memory for event sequences. This is an understudied area and this present work can make a major theoretical contribution to this domain with new predictions. The reviewers were positive about the contribution in review, and the revisions have clarified the model in a number of important ways, including through some additional simulation and analysis.

---

## [Decision Letter]

**Decision letter after peer review:**

Thank you for submitting your article ‘When to retrieve and encode episodic memories: a neural network model of hippocampal-cortical interaction’ for consideration by *eLife*. Your article has been reviewed by 2 peer reviewers, and the evaluation has been overseen by David Badre as Reviewing Editor and Michael Frank as the Senior Editor. The following individuals involved in review of your submission have agreed to reveal their identity: Tim Rogers (Reviewer #2); Michael E. Hasselmo (Reviewer #3).

Essential revisions:

The reviewers and editors were in agreement that this paper addresses an important and novel problem using a elegant but powerful approach. There was consensus that this can make a contribution to the literature, pending some revisions for clarify. The major points of clarification raised by the reviewers are provided below.

1) It is not clear how the sequences were constructed and how these relate to the structure of real schematic events. From the methods it looks like the inputs to the model contain a one-hot label indicating the feature observed (e.g., weather), a second one-hot vector indicating the feature value (e.g., rainy), and a third for the query, which indicates what prediction must be made (e.g., ‘what will the mood be’?). Targets then consist of potential answers to the various queries plus the ‘I don't know’ unit. If this is correct, two things are unclear. First, how does the target output (ie the correct prediction) relate to the prior and current features of the event? Like, is the mood always ‘angry’ when the feature is ‘rainy,’ or is it chosen randomly for each event, or does it depend on prior states? Does the drink ordered depend on whether the day is a weekday or weekend---in which case the LSTM is obviously critical since this occurs much earlier in the sequence---or does it only depend on the observed features of the current moment (e.g., the homework is a paper), in which case it's less clear why an LSTM is needed. Second, the details of the cover story (going to a coffee shop) didn't help to resolve these queries; for instance, Figure 2 seems to suggest that the kind of drink ordered depends on the nature of the homework assigned, which doesn't really make sense and makes reviewers question their understanding of Figure 2. In general this figure, example, and explanation of the model training/testing sequences could be substantially clarified. Maybe a figure showing the full sequence of inputs and outputs for one event, or a transition-probability graph showing how such patterns are generated, would be of help. Some further plain-language exposition of the cover-story and how it relates to the specific features, queries, and predictions shown in Figure 2 could be helpful.

2) The authors show that their model does a better job of using episodic traces to reinstate the event representation when such traces are stored at the end of an event, rather than when they are stored at both the middle and the end. In explaining why, they show that the model tends to organize its internal representations mainly by time (ie events occurring at a similar point along the sequence are represented as similar). If episodes for the middle of the event are stored, these are preferentially reinstated later as the event continues following distraction, since the start of the ‘continuing’ event looks more like an early-in-time event than a late-in-time event. This analysis is interesting and provides a clear explanation of why the model behaves as it does. However, reviewers want to know if it is mainly due to the highly sequential nature of the events the model is trained on, where prior observed features/queries don't repeat in an event. They wonder if the same phenomenon would be observed with richer sequences such as the coffee/tea-making examples from Botvinick et al., where one stirs the coffee twice (once after sugar, once after cream) and so must remember, for each stirring event, whether it is the first or the second. Does the ’encode only at the end’ phenomenon still persist in such cases? The result is less plausible as an account of the empirical phenomena if it only "works" for strictly linear sequences.

3) It would be helpful to understand how/why reinforcement learning is preferable to error-correcting learning in this framework. Since the task is simply to predict the upcoming answer to a query given a prior sequence of observed states, it seems likely that plain old backprop could work fine (indeed, the cortical model is pre-trained with backprop already), and that the model could still learn the critical episodic memory gating. Is the reinforcement learning approached used mainly so the model can be connected to the over-arching resource-rational perspective on cognition, or is there some other reason why this approach is preferred?

4) The details of the simulations provided in the methods section could be clarified. First, on page 20, it is unclear about how w_i is computed. It seems to compute w_i (activation of trace i) you need to first compute activation of all other traces j, but since all such computations depend each other it is not clear how this is carried out. Perhaps you initialize this values for tau=0, but some detail here would help. Second, the specification of the learning algorithm for the A2C objective on page 21 has some terms that aren't explained. Specifically, it is not clear what the capital J denotes and what the /pi denotes. Third, other details about the training sufficient to replicate the work should also be provided--for instance, an explanation of the "entropy regularization" procedure, learning rates, optimizer, etc, for the supervised pre-training, and so on.

5) Line 566 – "new frontier in the cognitive modeling of memory." They should qualify this statement with discussion of the shortcomings of these types of models. This model is useful for illustrating the potential functional utility of controlling the timing of episodic memory encoding and retrieval. However, the neural mechanisms for this control of gating is not made clear in these types of models. The authors suggest a portion of a potential mechanism when they mention the potential influence of the pattern of activity in the decision layer on the episodic memory gating (i.e., depending on the level of certainty – line 325), but they should mention that detailed neural circuit mechanisms are not made clear by this type of continuous firing rate model trained by gradient descent. More discussion of the difficulties of relating these types of neural network models to real biological networks is warranted. This is touched upon in lines 713-728, but the shortcomings of the current model in addressing biological mechanisms are not sufficiently highlighted. In particular, more biologically detailed models have addressed how network dynamics could regulate the levels of acetylcholine to shift dynamics between encoding and retrieval (Hasselmo et al., 1995; Hasselmo and Wyble, 1997). There should be more discussion of this prior work.

6) Figure 1 – "the cortical part of the model" – Line 117 – "cortical weights" and many other references to cortex. The hippocampus is a cortical structure (allocortex), so it is very confusing to see references to cortex used in contrast to hippocampus. The confusing use of references to cortex could be corrected by changing all the uses of "cortex" or "cortical" in this paper to "neocortex" or "neocortical."

7) "the encoding policy is not learned" – This came as a surprise in the discussion, so it indicates that this is not made sufficiently clear earlier in the text. This statement should be made in a couple of points earlier where the encoding policy is discussed.

---

## [Author Response]

1) It is not clear how the sequences were constructed and how these relate to the structure of real schematic events. From the methods it looks like the inputs to the model contain a one-hot label indicating the feature observed (e.g., weather), a second one-hot vector indicating the feature value (e.g., rainy), and a third for the query, which indicates what prediction must be made (e.g., "what will the mood be"?). Targets then consist of potential answers to the various queries plus the "I don't know" unit. If this is correct, two things are unclear. First, how does the target output (ie the correct prediction) relate to the prior and current features of the event? Like, is the mood always "angry" when the feature is "rainy," or is it chosen randomly for each event, or does it depend on prior states? Does the drink ordered depend on whether the day is a weekday or weekend---in which case the LSTM is obviously critical since this occurs much earlier in the sequence---or does it only depend on the observed features of the current moment (e.g., the homework is a paper), in which case it's less clear why an LSTM is needed. Second, the details of the cover story (going to a coffee shop) didn't help to resolve these queries; for instance, Figure 2 seems to suggest that the kind of drink ordered depends on the nature of the homework assigned, which doesn't really make sense and makes reviewers question their understanding of Figure 2. In general this figure, example, and explanation of the model training/testing sequences could be substantially clarified. Maybe a figure showing the full sequence of inputs and outputs for one event, or a transition-probability graph showing how such patterns are generated, would be of help. Some further plain-language exposition of the cover-story and how it relates to the specific features, queries, and predictions shown in Figure 2 could be helpful.

We are grateful to the reviewers for pointing out the need for further clarification of how we constructed the stimulus sequences. To answer the questions above: The target output at each time step is deterministically controlled by a particular feature of the situation. For example, the Barista state (Bob or Eve) that occurs at time step 1 is controlled by the Day situation feature – so, P(Barista = Bob |Day = weekday) = 1, and P(Barista = Eve |Day = weekend) = 1; the Mood state (happy or angry) that occurs at time step 2 is controlled by the Weather situation feature, and so on. The order of queries is fixed (the model is always asked about the Barista state on time step 1, the Mood state on time step 2, and so on) but – crucially – the order in which the situation features (Day, Weather, etc) are observed within an event is random. This combination of fixed-order queries but randomly-ordered situation feature observations implies that three possible circumstances can arise: (1) If the situation feature that controls a particular state is observed on the same time step that the state is queried, the model can respond directly without having to use memory. (2) If the situation feature that controls a particular state was observed earlier in the event, then the model can respond correctly by maintaining that feature in working memory. (3) If the situation feature that controls a particular state was not observed recently, the model can not use working memory to respond correctly, but it might be able to respond correctly if it retrieves an episodic memory of this situation.

In the previous draft, this information was spelled out later in the paper (to some extent in our exposition of Figure 2C, and more thoroughly in the “Stimulus representation” subsection of the Methods), but we agree with the reviewers that it would be more clear if we spelled this out earlier, when we discuss Figure 2A. Toward this end, we have made several changes to the paper:

First, we updated Figure 2A. We agree with the reviewer that having the Coffee state be controlled by the Homework feature was a bit obscure; we have restructured the example so the Drink that is ordered (juice or latte) is controlled by the Sleepiness feature (whether the protagonist is not sleepy or sleepy). We also modified the figure to address the three circumstances listed above (whether the relevant situation feature is observed before, during, or after the associated query, and the implications of each scenario for which memory systems are relevant).

We have also updated the caption and the main text to clarify the points made above. :

1. The caption of Figure 2:

“Figure 2. A situation-dependent event processing task. (A) An event is a sequence of states, sampled from an event schema and conditioned on a situation. An event schema is a graph where each node is a state. A situation is a collection of features (e.g., Day, Weather, Sleepiness) set to particular values (e.g., Day = weekday). The features of the current situation deterministically control how the event unfolds (e.g., the value of the Day feature controls which Barista state is observed). At each time point, the network observes the value of a randomly selected feature of the current situation, and responds to a query about what will happen next. Note that the order of queries is fixed but the order in which situation features are observed is random. If a situation feature (Sleepiness) is observed before the state it controls (Drink), the model can answer the query about that state by holding the relevant feature in working memory. However, if the relevant situation feature (Weather) is observed after the state it controls (Mood), the model can not rely on working memory on its own to answer the query. (B) We created three task conditions to simulate the design used by Chen et al., (2016): recent memory (RM), distant memory (DM), and no memory (NM); see text for details. (C) Decoded contents of the model’s working memory for an example trial from the DM condition. Green boxes indicate time points where the value of a particular situation feature was observed. The color of a square indicates whether the correct (i.e., observed) value of that feature can be decoded from the model’s working memory state (white = feature accurately decoded; black = feature not decoded). See text for additional explanation.”

2. Main text:

“To simulate the task of event processing, we define an event as a sequence of states,

sampled from an underlying graph that represents the event schema. […] If the relevant situation feature (Weather) for controlling a state (Mood) was not recently observed, the model can not use working memory on its own to answer the query; here, the only way the model can respond correctly is by retrieving a stored episodic memory of that situation.”

2) The authors show that their model does a better job of using episodic traces to reinstate the event representation when such traces are stored at the end of an event, rather than when they are stored at both the middle and the end. In explaining why, they show that the model tends to organize its internal representations mainly by time (ie events occurring at a similar point along the sequence are represented as similar). If episodes for the middle of the event are stored, these are preferentially reinstated later as the event continues following distraction, since the start of the "continuing" event looks more like an early-in-time event than a late-in-time event. This analysis is interesting and provides a clear explanation of why the model behaves as it does. However, reviewers want to know if it is mainly due to the highly sequential nature of the events the model is trained on, where prior observed features/queries don't repeat in an event. They wonder if the same phenomenon would be observed with richer sequences such as the coffee/tea-making examples from Botvinick et al., where one stirs the coffee twice (once after sugar, once after cream) and so must remember, for each stirring event, whether it is the first or the second. Does the "encode only at the end" phenomenon still persist in such cases? The result is less plausible as an account of the empirical phenomena if it only "works" for strictly linear sequences.

Thanks for the suggestion! We agree that this is a very interesting case. To respond to this suggestion, we ran a new simulation – as described in more detail below, we found that the “encode only at the end” phenomenon (i.e., better prediction performance for “encode only at the end” vs. “encode at end and also midway through”) persists even when the mapping between cues and responses is context-sensitive (i.e., the correct response to the first occurrence of a query in the sequence is different from the correct response to the second occurrence of that query).

In our standard task, each state is controlled by one (and only one) situation feature (e.g., weather, and only weather, controls mood). In our new simulation, each state (e.g., mood) has to be predicted twice in the sequence, and it is controlled by a different situation feature the first time that it occurs vs. the second time (e.g., mood is controlled by weather the first time it is queried in the sequence, and mood is controlled by music the second time it is queried in the sequence). As such, the model has to keep track of where it is in the sequence (i.e., is it being queried about mood for the first time or the second time) to know how to respond – in this key respect, the new simulation resembles the Botvinick and Plaut coffee-making example mentioned by the reviewer.

Specifically, for each sequence, the model was queried on 8 states that were controlled by 8 features, and then it was queried on the same 8 states again, but this time they were controlled by a different 8 features. In comparison, in our standard task, we had 16 distinct features that uniquely controlled 16 distinct states.

We found that the model can still learn to predict upcoming states very well with the new stimulus design. The model’s prediction performance and episodic memory gating pattern are qualitatively the same as what we found in the standard task (compare Author response image 1 to Figure 3 in the paper).

**Author response image 1. sa2fig1:** The model performed well in the task with repeating queries. This figure can be compared with Figure 3 in the paper.

Next, we examined how the new variant of the model performed when memories were encoded at the end of part 1 (only) vs. encoding at the end and also midway through. In the new simulation, as in our previous simulation, the (less informative) midway memory competes at retrieval with the (more informative) endpoint memory, interfering with retrieval of the endpoint memory and impairing prediction performance. The results of the simulation are shown in Author response image 2 (compare to Figure 4 in the paper).

**Author response image 2. sa2fig2:** Encoding midway through the event (in addition to encoding at the end) still hurts performance in the task with repeating queries. This figure can be compared with Figure 4 in the paper.

There are some subtle differences in the results obtained in the new simulation vs. the previous one: The fact that the same 8 queries are repeated in the first and second halves of the sequence makes the midway and endpoint memories more similar, so they are more evenly matched at retrieval (instead of the midway memory dominating the endpoint memory); because of the increased complexity of the mapping that the model has to learn, retrieving the midway memory blended together with the endpoint memory causes massive interference and results in a larger performance decrement than we observed in the previous simulation. Setting aside these differences, the basic point is unchanged – the extra “midway” memory impairs performance.

Given that the results of the new simulation lead to the same basic conclusions as the results of the existing simulation in the paper, we think that it is not worth devoting the space in the paper to give a full reporting of the methods and results for the new simulation. Instead, we have modified the main text to make it clear that the results described in the main text generalize to more complex sequences, and we restate the key principle that accounts for these results (starts at line 509):

“A possible alternative explanation of the negative effects of midway encoding is that midway encoding was introduced when we tested the model’s performance but was not present during meta-training (i.e., when the model acquired its retrieval policy); as such, midway encoding can be viewed as “out of distribution” and may be harmful for this reason. To address this concern, we also ran a version of the model where memories were stored both midway and at the end of an event during meta-training, and it was still true that endpoint-only encoding led to better performance than midway-plus-endpoint encoding; this result shows that midway encoding is intrinsically harmful, and it is not just a matter of it being out-of-distribution. In another simulation, we also found that the harmful effect of encoding midway through the sequence qualitatively replicates with more complex event structures (analogous to those in Botvinick and Plaut, 2004) where queries are repeated within a sequence and the model has to give a different response to the first vs. second occurrence of a query (e.g., mood is controlled first by weather and later by music).

To summarize the results from this section, the model does better when we force it to wait until the end of an event to take a snapshot. This pattern arises in our simulations because knowledge of the ongoing situation builds within an event, so memories encoded earlier in the event contain less information about the situation than memories encoded at the end. These less-informative midway memories harm performance by interfering with recall of more-informative endpoint memories, and are themselves more prone to false recall (because they are more confusable across events). Taken together, these simulation results provide a resource-rational justification for the results cited above showing preferential encodingrelated hippocampal activity at the end of events (Ben-Yakov and Dudai, 2011; Ben-Yakov et al., 2013; Baldassano et al., 2017; Ben-Yakov and Henson, 2018; Reagh et al., 2020).”

3) It would be helpful to understand how/why reinforcement learning is preferable to error-correcting learning in this framework. Since the task is simply to predict the upcoming answer to a query given a prior sequence of observed states, it seems likely that plain old backprop could work fine (indeed, the cortical model is pre-trained with backprop already), and that the model could still learn the critical episodic memory gating. Is the reinforcement learning approached used mainly so the model can be connected to the over-arching resource-rational perspective on cognition, or is there some other reason why this approach is preferred?

Thanks for pointing out this issue! We opted to use RL instead of purely supervised learning because of the model’s inclusion of a “don’t know” action. We expected that the optimal way to deploy this action would vary in complex ways as a function of different parameters (e.g., the penalty on incorrect responding), and we did not want to presume that we knew what the best policy would be. Consequently, we opted to let the model learn its own policy for using the “don’t know” action, rather than imposing a policy through supervised training. We now emphasize this point in the Model training section of the Methods.

A closely-related question is how our main model findings depend on the use of RL as opposed to supervised learning. The question is answered in Appendix 5, which shows results from a variant of the model that omitted the “don’t know” action and was trained with purely supervised learning. These simulations show that the primary effect of including the “don’t know” action is to make the model more patient (i.e. with “don’t know”, the model waits longer to initiate episodic retrieval, because the cost of delaying retrieval and saying “don’t know” is less than the cost of retrieving too early and giving the wrong answer) – apart from this, the results are quite similar. We now point the reader to Appendix 5 in the methods (as well as several other places in the text).

The changes we made to the Methods section are below:

“The model is trained with reinforcement learning. Specifically, the model is rewarded/penalized if its prediction about the next state is correct/incorrect. The model also has the option of saying “don’t know” (implemented as a dedicated output unit) when it is uncertain about what will happen next; if the model says “don’t know”, the reward is zero. The inclusion of this “don’t know” unit is what motivated us to use reinforcement learning as opposed to purely supervised training. We expected that the optimal way to deploy this action would vary in complex ways as a function of different parameters (e.g., the penalty on incorrect responding), and we did not want to presume that we knew what the best policy would be. Consequently, we opted to let themodel learn its own policy for using the “don’t know” action, rather than imposing a policy through supervised training (see Appendix 5 for results from a variant of the model where we omit the “don’t know” unit and train the model in a purely supervised fashion; these changes make the model less patient – i.e., retrieval takes place earlier in the sequence – but otherwise the results are qualitatively unchanged).

... Additionally, the “don’t know” output unit is not trained during the supervised pretraining phase – as noted above, we did this because we want the model to learn its own policy for saying “don’t know”, rather than having one imposed by us.”

4) The details of the simulations provided in the methods section could be clarified. First, on page 20, it is unclear about how w_i is computed. It seems to compute w_i (activation of trace i) you need to first compute activation of all other traces j, but since all such computations depend each other it is not clear how this is carried out. Perhaps you initialize this values for tau=0, but some detail here would help. Second, the specification of the learning algorithm for the A2C objective on page 21 has some terms that aren't explained. Specifically, it is not clear what the capital J denotes and what the /pi denotes. Third, other details about the training sufficient to replicate the work should also be provided--for instance, an explanation of the "entropy regularization" procedure, learning rates, optimizer, etc, for the supervised pre-training, and so on.

Thanks for pointing out that the description of the algorithm was unclear! We have now added more details in the Methods section: (1) We now describe the initial condition of w_i_, which is indeed zero for all i; (2) We added an explanation of the A2C objective, including the definitions of J andπ, and the formula for entropy regularization; And (3) we added information about the learning rate and optimizer to the method section. The changes that we made to the Methods section are marked in bold below (quoted passage starts at line 873):

*“w^i^_τ_* is the activation value for the i-th memory at time τ; these activations are set to zero initially (*w^i^*_0_ = 0, for all i). The activation for the *i*-th memory is positively relatedto its evidence, *x_i_*, and is multiplicatively modulated by α, the EM gate value”

(quoted passage resumes at line 899)

“The model is trained with the advantage actor-critic (A2C) objective (Mnih et al., 2016). At time *t*, the model outputs its prediction about the next state,. ŝ*_t+1_*, and an estimate of the state value, *v_t_*. After every event (i.e., a sequence of states of length *T*), it takes the following policy gradient step to adjust the connection weights for all layers, denoted by θ:∇θJ(θ)=∇[∑t=0Tlog⁡πθ(s^t+1|st)(rt−vt)−ηH(πθ(s^t+1|st))]

Where J is the objective function with respect to θ(the network weights). π_θ_ is the policy represented by the network weights. H is the entropy function that takes a probability distribution and returns its entropy η controls the strength of entropy regularization.

The objective function J makes rewarded actions (next-state predictions) more likely to occur; the above equation shows how this process is modulated by the level of reward

prediction error – measured as the difference between the predicted value, vt, versus the reward at time t, denoted by *r_t_*. We also used entropy regularization on the network output (Grandvalet and Bengio, 2006; Mnih et al., 2016) to encourage exploration in the early phase of the training process – with a positive η, the objective will encourage weight changes that lead to higher entropy distributions over actions. We used the A2C method (Mnih et al., 2016) because it is simple and has been widely used in cognitive modeling (Ritter et al., 2018; Wang et al., 2018). Notably, there is also evidence that an actor-critic style system is implemented in the neocortex and basal ganglia (Takahashi et al., 2008). Since pure reinforcement learning is not dataefficient enough, we used supervised initialization during meta-training to help the model develop useful representations (Misra et al., 2017; Naga- bandi et al., 2017). Specifically, the model is first trained for 600 epochs to predict the next state and to minimize the cross-entropy loss between the output and the target. During this supervised pre-training phase, the model is only trained on the recent memory condition and the episodic memory module is turned off, so this supervised pre-training does not influence the network’s retrieval policy. Additionally, the “don’t know” output unit is not trained during the supervised pre- training phase – as noted above, we did this because we want the model to learn its own policy for saying “don’t know”, rather than having one imposed by us. Next, the model is switched to the advantage actor-critic (A2C) objective (Mnih et al., 2016) and trained for another 400 epochs, allowing all weights to be adjusted. The number of training epochs was picked to ensure the learn- ing curves converge. For both the supervised-pretraining phase and the reinforcement learning phase, we used the Adam optimizer (Kingma and Ba, 2014) with learning rate of 7e-4 (for more details, please refer to Appendix 7).”

5) Line 566 – "new frontier in the cognitive modeling of memory." They should qualify this statement with discussion of the shortcomings of these types of models. This model is useful for illustrating the potential functional utility of controlling the timing of episodic memory encoding and retrieval. However, the neural mechanisms for this control of gating is not made clear in these types of models. The authors suggest a portion of a potential mechanism when they mention the potential influence of the pattern of activity in the decision layer on the episodic memory gating (i.e., depending on the level of certainty – line 325), but they should mention that detailed neural circuit mechanisms are not made clear by this type of continuous firing rate model trained by gradient descent. More discussion of the difficulties of relating these types of neural network models to real biological networks is warranted. This is touched upon in lines 713-728, but the shortcomings of the current model in addressing biological mechanisms are not sufficiently highlighted. In particular, more biologically detailed models have addressed how network dynamics could regulate the levels of acetylcholine to shift dynamics between encoding and retrieval (Hasselmo et al., 1995; Hasselmo and Wyble, 1997). There should be more discussion of this prior work.

Thanks for the suggestion! We updated the corresponding paragraph which is copied below (quoted passage starts at line 748):

“Going forward, we also hope to explore more biologically-realistic episodic memory models (e.g., Hasselmo and Wyble, 1997; Schapiro et al., 2017; Norman and O’Reilly, 2003; Ketz et al., 2013). […] We have opted to start with the simplified episodic memory system described in this paper both for reasons of scientific parsimony and also for practical reasons – adding additional neurobiological details would make the model run too slowly (the current model takes on the order of hours to run on standard computers; adding more complexity would shift this to days or weeks).”

6) Figure 1 – "the cortical part of the model" – Line 117 – "cortical weights" and many other references to cortex. The hippocampus is a cortical structure (allocortex), so it is very confusing to see references to cortex used in contrast to hippocampus. The confusing use of references to cortex could be corrected by changing all the uses of "cortex" or "cortical" in this paper to "neocortex" or "neocortical."

Thanks for the suggestion! We changed cortical/cortex to neocortex/neocortical throughout the paper.

7) "the encoding policy is not learned" – This came as a surprise in the discussion, so it indicates that this is not made sufficiently clear earlier in the text. This statement should be made in a couple of points earlier where the encoding policy is discussed.

Thanks for pointing out that this is unclear! Now we mention this explicitly at the beginning of the encoding simulation (quoted passage starts at line 473):

“While it is clear why encoding at the end of an event is useful, it is less clear why encoding at other times might be harmful; naively, one might think that storing more episodic snapshots during an event would lead to better memory for the event. To answer this question, we compared models that selectively encode episodic memories at the end of each event to models that encode episodic memories both at the end of each event and also midway through each event. Note that the model did not learn these two encoding policies by itself – we simply configured the model to encode one way or the other and compared their performance. If selectively encoding at the end of an event yields better performance, this would provide a resource-rational justification for the empirical findings reviewed above.”